# Radial Scaling Voxelization for Accurate Small Object 3D Detection

**Hao Liu** [1 2 *]  **Yi Zhou** [1]  **Yanni Ma** [3 *]

## Abstract

Voxel-based 3D object detectors typically discretize the spatial domain using a uniform Cartesian grid, which allocates the same voxel size to both near-range and far-range regions. However, this uniform discretization is suboptimal for small objects such as pedestrians and cyclists, as they occupy only a few voxels and thus struggle to capture fine-grained geometric details. Although increasing the global voxel resolution can alleviate this problem, it inevitably increases substantial memory consumption and computational cost. In this paper, we propose Radial Scaling Voxelization (RSV), a simple yet effective non-uniform discretization strategy that adaptively modulates the effective voxel size based on the radial distance from the LiDAR sensor. Unlike previous cylindrical or polar discretization schemes, RSV preserves the Cartesian grid topology by applying a continuous radial scaling function to the input coordinates before standard voxelization. This operation yields a near-high, far-unchanged resolution pattern, i.e., the effective voxel size becomes finer in near regions, where the geometric structures of small objects are difficult to capture, while remaining nearly unchanged in far regions to avoid unnecessary computational cost. Importantly, RSV is architecture-agnostic and can directly replace the discretization module in any voxel-based detector without modifying the backbone, network design, or training pipeline. Extensive experiments on the KITTI and nuScenes datasets demonstrate that integrating our RSV into several voxel-based baselines consistently enhances small-object detection per-

formance, especially for the Pedestrian and Cyclist categories, while incurring only marginal additional computational overhead. Code is available at `https://github.com/Zeoy2020/RadialScalingVoxelization`.

## 1. Introduction

Accurate 3D object detection from LiDAR point clouds is a fundamental problem in computer vision with broad applications, such as autonomous driving, robotics, and drone navigation. The goal of 3D object detection is to localize objects of interest and recognize their categories, which is essential for enabling machines to understand and interact with their surrounding environments. In recent years, several approaches have been proposed to tackle this problem, each with its own strengths and limitations. Among these, voxel-based detectors have received significant attention for their ability to balance detection accuracy and computational efficiency. By discretizing irregular point clouds into regular cells, these methods enable the efficient use of (sparse) 3D convolutions, which facilitates scalable and high-performance object detection. It is worth noting that the discretization process plays a crucial role in determining the geometric fidelity of the resulting feature representation.

Current voxel-based methods typically rely on uniform discretization in a Cartesian coordinate system, which assigns the same voxel size to both near-range and far-range regions. However, uniform discretization leads to suboptimal performance for small objects, such as pedestrians and cyclists, as they occupy only a few voxels and thus their fine-grained geometric structures are severely underrepresented. A straightforward solution is to increase the voxel resolution. As shown in Fig. 1, reducing the voxel size from $0.1m$ to $0.05m$ brings significant performance improvements for both pedestrians and cyclists. Specifically, for pedestrians, detection performance is improved by 14.63 AP points overall and 13.96 AP points at the 0-30$m$ range, with similar improvements observed for cyclists. Note that, at the $>50$ m range, reducing the voxel size does not improve performance, as fewer points are present in these regions, and finer voxelization may cause object voxels to be incorrectly treated as noise. These performance gains come at significant memory and computational cost, making this solution

[1]School of Geospatial Artificial Intelligence, East China Normal University, Shanghai, China [2]Key Laboratory of Geographic Information Science (Ministry of Education), East China Normal University, Shanghai, China [3]School of Electronics and Communication Engineering, Sun Yat-sen University, Shenzhen, China. Correspondence to: Hao Liu <hliu@geoai.ecnu.edu.cn>, Yanni Ma <mayn3@mail2.sysu.edu.cn>.

*Proceedings of the 43rd International Conference on Machine Learning*, Seoul, South Korea. PMLR 306, 2026. Copyright 2026 by the author(s).

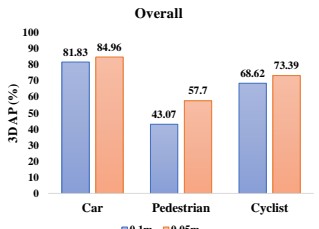 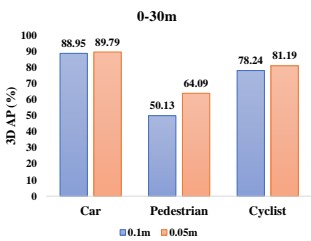 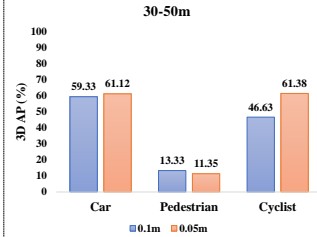 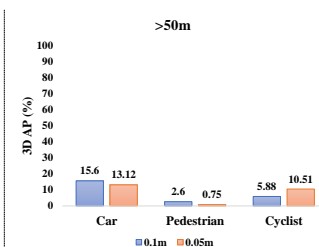

*Figure 1.* Impact of voxel resolution on the 3D detection performance on the KITTI dataset. We report 3D AP of a voxel-based detector Voxel-RCNN for Car, Pedestrian, and Cyclist using a coarse voxel size of $0.1m$ (blue) and a fine voxel size of $0.05m$ (orange), evaluated over (a) Overall, (b) 0–30$m$, (c) 30–50$m$, and (d) >50$m$. Increasing voxel resolution yields only marginal gains for Car across all distance ranges, but leads to clear improvements for Pedestrian and Cyclist in the near and mid ranges (0–30$m$ and 30–50$m$), while the far range shows limited benefit. This observation motivates our RSV design, which allocates finer voxel resolution to near-range regions while keeping far-range resolution largely unchanged to preserve efficiency.

unsuitable for practical applications.

Several approaches (Zhu et al., 2021; Wang et al., 2025; 2021; He et al., 2022) have been proposed to overcome the limitations of uniform discretization. For example, cylindrical or polar coordinate systems inspire non-uniform voxel partitioning (Zhu et al., 2021; Wang et al., 2025), but they disrupt the Cartesian topology commonly used in mainstream 3D backbones. This disruption complicates feature processing and prevents the direct integration of existing voxel-based architectures. Other adaptive discretization strategies (Wang et al., 2021; He et al., 2022) typically require significant modifications to the network design or rely on heuristic grid shapes that may not align well with the LiDAR sensing pattern. Overall, existing approaches fail to provide a simple, plug-and-play solution that significantly improves small-object detection performance without modifying network design or introducing heavy memory and computational overhead.

In this paper, we propose Radial Scaling Voxelization (RSV), a non-uniform discretization strategy that adaptively adjusts the effective voxel size based on the radial distance from the LiDAR sensor. Specifically, RSV allocates finer voxel resolution to near-range regions to better preserve geometric details, while using coarser voxels in far-range areas to reduce memory and computational cost. By redistributing discretization capacity in this way, RSV improves small-object detection performance without introducing substantial memory and computational overhead, with particularly significant gains for near-range small objects. More importantly, our RSV preserves the Cartesian grid topology, making it a plug-and-play replacement for standard voxelization and enabling seamless integration with existing voxel-based detectors. The main contributions of this paper are summarized as follows:

- We demonstrate that small-object detection performance is positively correlated with voxel resolution, revealing an inherent trade-off in uniform discretiza-

tion: using small voxels greatly increases computational cost, while using large voxels fails to capture the fine-grained geometric information of small objects.

- We propose Radial Scaling Voxelization, an effective non-uniform discretization strategy that adaptively adjusts the effective voxel size based on the radial distance from the LiDAR sensor.

- We show that RSV is architecture-agnostic and can be seamlessly integrated into existing voxel-based detectors. Experimental results on KITTI and nuScenes show consistent improvements, particularly for small objects such as pedestrians and cyclists.

**Conflict of Interest Disclosure.** The authors declare no financial or other substantive conflicts of interest that could reasonably be perceived to influence this work.

## 2. Related Work

### 2.1. Voxel-based 3D Object Detection

3D object detection aims to localize objects of interest in a 3D scene. According to their point cloud processing strategy, existing methods are broadly classified into projection-based, point-based, and voxel-based approaches. Projection-based methods (Meyer et al., 2019b;a; Fan et al., 2021; Bai et al., 2023) project irregular 3D point clouds into structured 2D representations such as bird's eye view (BEV), front view, and spherical range images, allowing the detector to leverage mature 2D pipelines. Point-based methods (Shi et al., 2019; Yang et al., 2020; Zhang et al., 2022; Liu et al., 2023a) typically employ point-wise Multi-layer Perceptron (MLP) and local neighborhood aggregation to learn fine-grained geometric details directly from raw point clouds. Although these methods can achieve high detection accuracy, they fail to handle large-scale point cloud data.

Voxel-based methods (Yan et al., 2018; Lang et al., 2019; Deng et al., 2021; Wu et al., 2023; Yu et al., 2025) discretize

the 3D space into regular cells and apply 3D (sparse) convolutions for feature learning. Although discretization may lose fine-grained geometry, these methods strike a favorable balance between accuracy and efficiency. VoxelNet (Zhou & Tuzel, 2018) divides the point cloud into equally spaced 3D voxels and encodes them using a voxel feature encoding (VFE) layer to form a dense 4D tensor, followed by a region proposal network to generate 3D bounding boxes. Despite its impressive performance, the dense 3D convolutions introduce heavy computational burden. SECOND (Yan et al., 2018) addresses this limitation by introducing 3D sparse convolution, which significantly improves computational efficiency without sacrificing detection accuracy. Building on sparse voxel features, Voxel-RCNN (Deng et al., 2021) introduces a voxel RoI pooling module that enables accurate and efficient second-stage refinement. PointPillars (Lang et al., 2019) streamlines voxelization by partitioning points into vertical pillars. The resulting pillar features are squeezed into a 2D pseudo image and processed by a 2D detection network. CenterPoint (Yin et al., 2021) uses a voxel- or pillar-based encoder to learn 3D representations and projects them onto the BEV plane. The two anchor-free heads are then used to predict object centers and regress 3D box attributes such as size and heading angle. VirConv (Wu et al., 2023) introduces virtual point clouds to enrich voxel-based representations. To mitigate the redundancy and noise introduced by these virtual points, it adopts a Stochastic Voxel Discard (StVD) module to remove excessive nearby voxels and a Noise-Resistant Submanifold Convolution (NRConv) to stabilize geometric feature learning. Recently, voxel-based detectors have evolved toward LiDAR–image fusion (Bai et al., 2022; Liu et al., 2023b; Cai et al., 2023; Liu et al., 2025), which leverages both geometric and visual cues to enhance detection performance. However, voxel-based detectors still struggle with small object detection due to the mismatch between voxel resolution and fine-scale geometry of small objects. Although increasing the voxel resolution can theoretically alleviate this issue, it leads to prohibitive memory and computational costs.

## 2.2. Discretization for 3D Object Detection

Discretization plays a crucial role in voxel-based 3D object detection, as it converts irregular point clouds into structured representations suitable for convolutional processing. Existing discretization strategies can be roughly categorized into uniform discretization and non-uniform discretization.

Uniform discretization methods divide the 3D space into regular voxels or vertical pillars. Their simplicity and compatibility with sparse convolutions make them widely adopted. However, fixed uniform grids suffer from a fundamental accuracy-efficiency trade-off: fine grids preserve geometric details but incur high computation and memory costs, while coarse grids are efficient but degrade small object

detection performance. To mitigate this issue, recent works (Wang et al., 2021; He et al., 2022; Zhai et al., 2024; Zhu et al., 2021) have increasingly explored non-uniform discretization. Wang et al. (Wang et al., 2021) proposed reconfigurable voxelization, which dynamically adjusts voxel shapes based on local point distributions to better accommodate the varying density of LiDAR points. He (He et al., 2022) modulated voxel size using image category and spatial depth clustering to allocate higher resolution to geometrically complex regions. Zhu et al. (Zhu et al., 2021) partitioned raw point clouds in cylindrical coordinates to align the voxelization with the scanning pattern of LiDAR sensors. Subsequently, Wang et al. (Wang et al., 2025) proposed a non-uniform cylindrical voxelization, which increases radial intervals with distance to match point density and improve efficiency. Although these methods effectively alleviate the non-uniform density of point clouds and demonstrate promising results in downstream tasks, they inevitably disrupt the Cartesian topology commonly adopted in mainstream 3D backbones. Such disruption complicates feature processing and undermines compatibility with sparse convolutions, making it difficult to directly integrate with existing voxel-based detectors without architectural modifications.

In this paper, we propose a non-uniform Radial Scaling Voxelization strategy that adaptively adjusts the effective voxel size according to the radial distance from the LiDAR sensor. This design preserves the Cartesian grid topology and is seamlessly integrated with existing voxel-based detectors, while significantly improving small object detection performance with only a marginal increase in computational and memory overhead.

## 3. Methodology

### 3.1. Overview

Discretization is a fundamental step in voxel-based 3D detectors, as it converts irregular point clouds into structured representations that can be processed by sparse 3D convolutional backbones. However, conventional uniform discretization employs a single, fixed voxel size for all spatial locations. This inevitably leads to a resource trade-off: coarse voxels are computationally affordable but lose fine-grained geometry, whereas globally increasing the voxel resolution becomes prohibitive in terms of memory and computation.

An empirical observation in Fig. 1 further substantiates this limitation. When the voxel size is reduced (e.g., from $0.1m$ to $0.05m$), the detection performance for small objects such as pedestrians and cyclists is significantly improved, with particularly notable gains in the 0-30$m$ range. In contrast, far-range regions, where LiDAR points are much sparser, do not exhibit comparable benefits from finer discretiza-

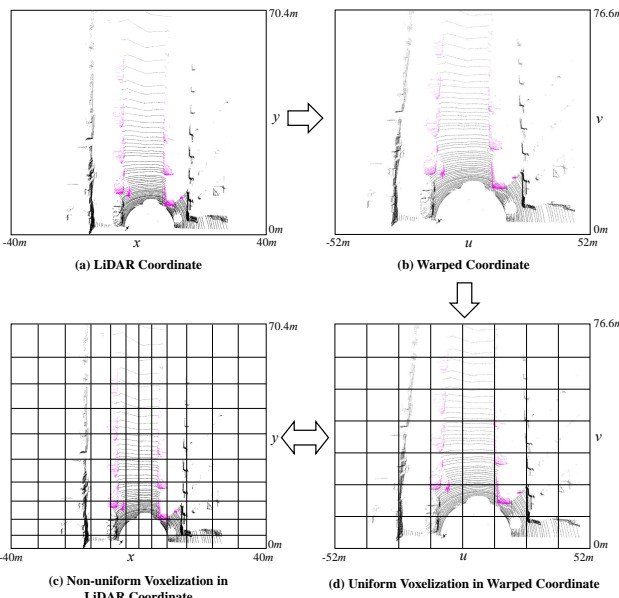

**(a) LiDAR Coordinate**

**(b) Warped Coordinate**

**(c) Non-uniform Voxelization in LiDAR Coordinate**

**(d) Uniform Voxelization in Warped Coordinate**

*Figure 2.* Overview of Radial Scaling Voxelization.

tion. These results suggest that the voxel resolution should be *distance-adaptive*, i.e., allocating finer voxels to near-range regions, while maintaining or even relaxing the voxel resolution in the far range to preserve efficiency.

Motivated by this, we propose Radial Scaling Voxelization (RSV), a non-uniform discretization strategy that adaptively adjusts the effective voxel size based on the radial distance from the LiDAR sensor. As shown in Fig. 2, RSV performs voxelization in a transformed coordinate space obtained through a distance-dependent nonlinear scaling of the horizontal plane. This transformation introduces strong expansion in the near range and gradually attenuates with increasing radial distance, resulting in finer voxel resolution close to the ego-vehicle while keeping the far-range resolution largely unchanged. More importantly, the transformation preserves the Cartesian grid topology, ensuring seamless integration with existing voxel-based detectors without modifying network architectures.

During both training and inference, we first warp the point cloud into the transformed space and then apply standard voxelization and sparse convolution as in existing voxel-based detectors. Ground-truth boxes are transformed into the same (transformed) space for supervision, and the predicted boxes are finally mapped back to the original LiDAR coordinates via the inverse transformation for evaluation.

### 3.2. Radial Scaling Voxelization

**Radial Scaling Transformation.** Given a 3D point $(x, y, z)$ in the LiDAR coordinate system, RSV introduces a smooth radial scaling to the horizontal plane:

$$(u, v, w) = (s(r)x, \ s(r)y, \ z), \tag{1}$$

where $r = \sqrt{x^2 + y^2}$ denotes the radial distance. The scaling factor $s(r)$ is defined as an exponential decay function:

$$s(r) = 1 + (s_{\max} - 1) \exp\left(-\beta \frac{r}{r_{\text{far}}}\right), \tag{2}$$

where $s_{\max} > 1$ controls the strength of near-range expansion, $r_{\text{far}}$ specifies the distance at which the scaling approaches 1, and $\beta$ determines the decay rate. This function satisfies two desirable properties:

- **Near-range expansion:** $s(0) = s_{\max}$, creating finer voxel resolution around the ego-vehicle.

- **Far-range stability:** $s(r) \to 1$ as $r \to r_{\text{far}}$, ensuring that the effective voxel resolution in far-range regions remains essentially unchanged.

Importantly, the transformation is monotonic and smooth in $r$, and it preserves the Cartesian grid topology because the mapping scales only the $x$ and $y$ axes without rotation or shearing. This property is crucial for maintaining compatibility with existing voxel-based network architectures.

**Inverse Radial Scaling Transformation.** Given a transformed point $(u, v, w)$, RSV recovers its corresponding Li-DAR coordinate $(x, y, z)$ by inverting the radial scaling applied on the horizontal plane. From the forward mapping in Eq. 1, the inversion cannot be derived in closed form because $s(r)$ depends on the unknown radius $r$. To address this, we solve for $r$ via the Newton-Raphson method (See Appendix A), which iteratively finds the root of:

$$f(r) = s(r)r - \rho = 0, \qquad \rho = \sqrt{u^2 + v^2}. \tag{3}$$

The smooth and strictly monotonic nature of $s(r)$ guarantees rapid and reliable convergence, typically within 5–8 iterations. Once $r$ is estimated, the inverse mapping is computed as:

$$x = \frac{r}{\rho}u, \qquad y = \frac{r}{\rho}v, \qquad z = w. \tag{4}$$

This inversion step is lightweight and numerically stable, introducing negligible reconstruction error. As a result, all transformed coordinates can be consistently mapped back to the original LiDAR coordinate system.

**Radial Scaling Voxelization.** RSV performs voxelization in the transformed $(u, v, w)$ space using a fixed voxel size $(\Delta u, \Delta v, \Delta w)$. When mapped back to the LiDAR coordinates, the effective voxel size becomes:

$$\Delta x = \frac{\Delta u}{s(r)}, \quad \Delta y = \frac{\Delta v}{s(r)}, \quad \Delta z = \Delta w. \tag{5}$$

As a result, the LiDAR space voxel resolution naturally adapts to the radial distance:

- **Near-range:** large $s(r) \rightarrow$ finer voxels, better capturing the geometry details of small objects.

- **Far-range:** $s(r) \approx 1$ keeping original voxel size and thus improving computational efficiency.

Because voxelization is conducted on the warped point clouds rather than altering the grid itself, RSV introduces no modification to voxel feature encoding, sparse convolution, or subsequent detection modules. Furthermore, RSV introduces no extra learnable parameters and incurs only a marginal increase in memory and computational overhead, but it brings significant improvements in small object detection performance for existing voxel-based detectors. The overall procedure of our RSV is summarized in Appendix B.

### 3.3. Integration with Existing Detectors

RSV is designed as a plug-and-play replacement for standard voxelization in voxel-based 3D detectors. It can be integrated into existing pipelines without modifying the backbone or detection head, i.e., the detector architecture remains unchanged and the original training / inference pipeline can be directly reused. The only difference is that voxelization and supervision are performed in the warped space rather than the original LiDAR coordinates. Accordingly, RSV integration involves three simple steps: (1) augmenting point-wise features with RSV-related attributes, (2) transforming 3D bounding boxes between the LiDAR space and the warped space, and (3) reusing the original detection losses by computing them on warped-space predictions and ground truth.

**Point-wise Feature Augmentation.** For each raw LiDAR point $\mathbf{p} = (x, y, z)$, we first compute its planar radial distance $r = \sqrt{x^2 + y^2}$ and the corresponding scaling factor $s(r)$. The point is then warped to the $(u, v, w)$ space according to Eq. 1. To make the detector aware of the distance-dependent warping, we augment the original point feature $(x, y, z, \text{intensity})$ with three RSV-related attributes: (i) the normalized radial distance $r_{\text{norm}}$, (ii) the logarithmically normalized radial distance $r_{\text{log\_norm}}$, and (iii) the min–max normalized logarithm of the scaling factor $s_{\text{log\_norm}}$. Let $[x_{\min}, y_{\min}, z_{\min}, x_{\max}, y_{\max}, z_{\max}]$ denote the predefined point cloud range. The maximum planar range is defined as:

$$r_{\max}^{pc} = \sqrt{(x_{\max} - x_{\min})^2 + (y_{\max} - y_{\min})^2}. \quad (6)$$

For each point, we compute

$$r_{\text{norm}} = \text{clip}\left(\frac{r}{r_{\max}^{pc} + \varepsilon}, 0, 1\right), \quad (7)$$

$$r_{\text{log\_norm}} = \frac{\log(1 + r)}{\log(1 + r_{\max}^{pc})}, \quad (8)$$

$$s_{\text{log\_norm}} = \frac{\log s(r) - \min_{\mathbf{p}} \log s(r)}{\max_{\mathbf{p}} \log s(r) - \min_{\mathbf{p}} \log s(r)}, \quad (9)$$

where $\min_{\mathbf{p}}(\cdot)$ and $\max_{\mathbf{p}}(\cdot)$ are taken over all points in the current point cloud. The resulting point-wise feature vector is $(u, v, w, \text{intensity}, r_{\text{norm}}, r_{\text{log\_norm}}, s_{\text{log\_norm}})$.

**Bounding Box Transformation and Inverse Mapping.** Since supervision is performed in the warped space, we convert ground-truth boxes from the LiDAR coordinates to the RSV-warped coordinates. Given a 3D bounding box parameterized by $(x_c, y_c, z_c, l_{\text{box}}, w_{\text{box}}, h_{\text{box}}, \theta)$ in the LiDAR space, we map its center $(x_c, y_c, z_c)$ to $(u_c, v_c, w_c)$ using Eq. 1. To transform the box size $(l_{\text{box}}, w_{\text{box}}, h_{\text{box}})$, we adopt a center-based approximation by evaluating the scaling factor at the center radial distance $r_c = \sqrt{x_c^2 + y_c^2}$:

$$(l'_{\text{box}}, w'_{\text{box}}, h'_{\text{box}}) = (s(r_c)l_{\text{box}}, s(r_c)w_{\text{box}}, h_{\text{box}}) \quad (10)$$

The yaw angle is kept unchanged, i.e., $\theta' = \theta$, since RSV introduces neither rotation nor shearing. At inference time, the detector outputs boxes in the warped space. We first map the predicted center back to the LiDAR space using the inverse transformation (Eq. 3-4), thus obtaining $(x_c, y_c, z_c)$. We then compute $r_c$ and apply the inverse scaling to recover box sizes, i.e., $l_{\text{box}} = l'_{\text{box}}/s(r_c)$ and $w_{\text{box}} = w'_{\text{box}}/s(r_c)$, while keeping $h_{\text{box}} = h'_{\text{box}}$ unchanged.

**Loss Computation.** RSV does not alter the loss design of existing detectors. We use the same classification and box regression losses as the baseline, but compute them between predictions and ground-truth boxes in the warped space. In other words, the detection model is trained to regress warped-space box parameters with the original loss formulations and weights, ensuring seamless reuse of existing training objectives.

## 4. Experiments

### 4.1. Experimental Settings

**Datasets.** We evaluate our method on the KITTI (Geiger et al., 2012) and nuScenes (Caesar et al., 2020) datasets. KITTI is one of the most widely used benchmarks for LiDAR-based 3D object detection in autonomous driving. It contains 7,481 annotated samples and 7,518 test samples. Following the official data split, the annotated set is further divided into a training set with 3,712 samples and a validation set with 3,769 samples. nuScenes is the first large-scale point cloud dataset for autonomous vehicles. It includes 1000 video sequences with a duration of 20 seconds, where 700 sequences for training, 150 sequences for validation and 150 sequences for testing.

**Evaluation Metric.** We use the Average Precision (AP) as the main evaluation metric. For the KITTI dataset, we report AP for the *Car*, *Pedestrian*, and *Cyclist* categories

*Table 1.* Comparative 3D object detection results on the KITTI validation set. Improvements brought by RSV (Δ gain) are shown in red.

| Method | Voxel Size (m) | Pedestrian AP$_{3D}$ (R40)↑ | | | | Cyclist AP$_{3D}$ (R40)↑ | | | | Car | Latency (ms) | FPS |
|---|---|---|---|---|---|---|---|---|---|---|---|---|
| | | Easy | Mod. | Hard | Overall | Easy | Mod. | Hard | Overall | | | |
| SECOND-IoU (Yan et al., 2018) | [0.1 , 0.1 , 0.1 ] | 49.43 | 42.37 | 37.70 | 43.16 | 78.45 | 60.42 | 56.41 | 65.09 | 78.85 | 50.00 | 20.00 |
| SECOND-IoU + RSV | [0.1 , 0.1 , 0.1 ] | 56.98 | 50.61 | 46.35 | 51.31 | 88.58 | 69.52 | 65.13 | 74.41 | 78.62 | 56.62 | 17.66 |
| Δ Gain | - | **+7.55** | **+8.24** | **+8.65** | **+8.15** | **+10.13** | **+9.10** | **+8.72** | **+9.32** | **-0.23** | - | - |
| SECOND-IoU (Yan et al., 2018) | [0.05, 0.05, 0.1 ] | 61.06 | 54.45 | 49.14 | 54.88 | 90.56 | 69.83 | 65.43 | 75.27 | 82.20 | 51.53 | 19.41 |
| SECOND-IoU + RSV | [0.05, 0.05, 0.1 ] | 61.89 | 55.34 | 50.45 | 55.89 | 91.02 | 72.49 | 68.10 | 77.20 | 81.92 | 57.64 | 17.35 |
| Δ Gain | - | **+0.83** | **+0.89** | **+1.31** | **+1.01** | **+0.46** | **+2.66** | **+2.67** | **+1.93** | **-0.28** | - | - |
| PointPillars (Lang et al., 2019) | [0.32, 0.32, 4.0 ] | 25.91 | 23.69 | 22.44 | 24.01 | 59.61 | 43.61 | 40.88 | 48.03 | 68.40 | 35.22 | 28.39 |
| PointPillars + RSV | [0.32, 0.32, 4.0 ] | 56.28 | 50.65 | 46.00 | 50.98 | 81.29 | 59.81 | 56.35 | 65.82 | 72.91 | 43.25 | 23.12 |
| Δ Gain | - | **+30.37** | **+26.96** | **+23.56** | **+26.97** | **+21.68** | **+16.20** | **+15.47** | **+17.79** | **+4.51** | - | - |
| PointPillars (Lang et al., 2019) | [0.16, 0.16, 4.0 ] | 54.90 | 49.92 | 45.50 | 50.11 | 79.47 | 62.61 | 58.81 | 66.96 | 78.70 | 38.08 | 26.26 |
| PointPillars + RSV | [0.16, 0.16, 4.0 ] | 57.06 | 51.88 | 48.83 | 52.59 | 86.54 | 66.73 | 61.91 | 71.73 | 78.71 | 41.04 | 24.36 |
| Δ Gain | - | **+2.16** | **+1.96** | **+3.33** | **+2.48** | **+7.07** | **+4.12** | **+3.10** | **+4.77** | **+0.01** | - | - |
| Voxel-RCNN (Deng et al., 2021) | [0.1 , 0.1 , 0.1 ] | 49.83 | 43.07 | 39.14 | 44.02 | 88.96 | 68.62 | 64.50 | 74.03 | 81.83 | 52.53 | 19.03 |
| Voxel-RCNN + RSV | [0.1 , 0.1 , 0.1 ] | 69.20 | 62.19 | 56.71 | 62.70 | 92.23 | 73.39 | 68.61 | 78.08 | 82.91 | 58.35 | 17.14 |
| Δ Gain | - | **+19.37** | **+19.12** | **+17.57** | **+18.68** | **+3.27** | **+4.77** | **+4.11** | **+4.05** | **+1.08** | - | - |
| Voxel-RCNN (Deng et al., 2021) | [0.05, 0.05, 0.1 ] | 65.02 | 57.70 | 52.59 | 58.44 | 90.85 | 74.05 | 69.12 | 78.01 | 84.96 | 58.55 | 17.08 |
| Voxel-RCNN + RSV | [0.05, 0.05, 0.1 ] | 70.31 | 63.92 | 58.20 | 64.14 | 93.30 | 73.81 | 69.10 | 78.74 | 85.20 | 58.82 | 17.00 |
| Δ Gain | - | **+5.29** | **+6.22** | **+5.61** | **+5.70** | **+2.45** | **–0.24** | **–0.02** | **+0.73** | **+0.24** | - | - |
| VirConv-L (Wu et al., 2023) | [0.1 , 0.1 , 0.05] | 56.25 | 47.41 | 42.06 | 48.57 | 84.36 | 62.02 | 57.56 | 67.98 | 84.57 | 128.02 | 7.81 |
| VirConv-L + RSV | [0.1 , 0.1 , 0.05] | 70.95 | 63.08 | 56.20 | 63.41 | 88.66 | 65.53 | 60.73 | 71.64 | 84.55 | 139.99 | 7.14 |
| Δ Gain | - | **+14.70** | **+15.67** | **+14.14** | **+14.84** | **+4.30** | **+3.51** | **+3.17** | **+3.66** | **-0.02** | - | - |
| VirConv-L (Wu et al., 2023) | [0.05, 0.05, 0.05] | 66.32 | 57.90 | 51.44 | 58.55 | 88.63 | 62.98 | 58.78 | 70.13 | 87.14 | 152.43 | 6.56 |
| VirConv-L + RSV | [0.05, 0.05, 0.05] | 69.97 | 62.46 | 55.53 | 62.65 | 88.44 | 62.98 | 58.84 | 70.09 | 86.24 | 158.40 | 6.31 |
| Δ Gain | - | **+3.65** | **+4.56** | **+4.09** | **+4.10** | **–0.19** | **+0.00** | **+0.06** | **–0.04** | **-0.90** | - | - |

under the Easy, Moderate, and Hard difficulty levels. The AP values are computed over 40 recall positions, with IoU thresholds of 0.7 for *Car* and 0.5 for *Pedestrian* and *Cyclist*. For the nuScenes dataset, AP is computed using center distance thresholds (i.e., 0.5m, 1.0m, 1.5m, and 2.0m) instead of IoU thresholds. The final mean AP (mAP) is obtained by averaging AP over these four distance thresholds. In addition, nuScenes introduces the nuScenes detection score (NDS) metric, which aggregates mAP and five true-positive quality terms. Concretely, NDS = $\frac{1}{10}\left[5 \times \text{mAP} + \sum_{m \in \{\text{ATE,ASE,AOE,AVE,AAE}\}} (1 - \min(1, m))\right]$, where ATE / ASE / AOE / AVE / AAE measure translation / scale / orientation / velocity / attribute errors.

**Implementation Details.** Our radial scaling voxelization (RSV) is designed as a drop-in replacement for the standard uniform voxelization used in existing voxel-based methods. We integrate RSV into four representative detectors, including SECOND (Yan et al., 2018), PointPillars (Lang et al., 2019), Voxel-RCNN (Deng et al., 2021), and VirConv (Wu et al., 2023).

In all cases, RSV only replaces the discretization step, and the remaining components such as sparse convolutional backbones, detection heads, training objectives, and post-processing pipelines remain completely unchanged. For all experiments, we adopt a consistent radial scaling configuration with $s_{\max} = 2$, $\beta = 1.5$, and $r_{\text{far}} = 50m$. These settings can significantly improve voxel resolution in near-range regions while keeping the far-range resolution essentially unchanged. During preprocessing, raw LiDAR points are first transformed by the radial scaling transformation and then voxelized in the transformed coordinate

space using the same voxel size as in the corresponding baseline. At inference time, predicted 3D bounding boxes are mapped back to the original LiDAR coordinate system using the inverse transformation. All baseline detectors follow their official training configurations, including learning rate schedule, optimizer, batch size, data augmentation, and training epochs. For fair comparisons, we do not make any changes to hyperparameters or training pipelines.

### 4.2. Main Results

**Results on the KITTI validation set.** We integrate RSV into four representative voxel-based detectors, including SECOND, PointPillars, Voxel-RCNN, and VirConv. Table 1 summarizes their detection results on the KITTI validation set. We observe that RSV consistently yields substantial improvements in small-object detection across all baselines, while retaining their performance on large objects such as cars. Specifically, cyclist overall AP increases by up to 17.79 points across different voxel settings, and pedestrian overall AP shows even larger gains of 1.01–26.97 points. Notably, the improvements are larger under coarser voxel resolutions. This is mainly due to the non-linear relationship between voxel resolution and detection performance, where simply increasing the resolution does not translate to proportional performance gains. More importantly, RSV does not introduce significant computational overhead, increasing inference latency by only 0.27–11.97 ms per sample. Table 2 presents the comparison of training memory usage. It can be observed that RSV leads to a moderate increase in peak GPU memory compared with the corresponding uniform baseline, but remains substantially more efficient than globally increasing the voxel resolution. This shows that RSV

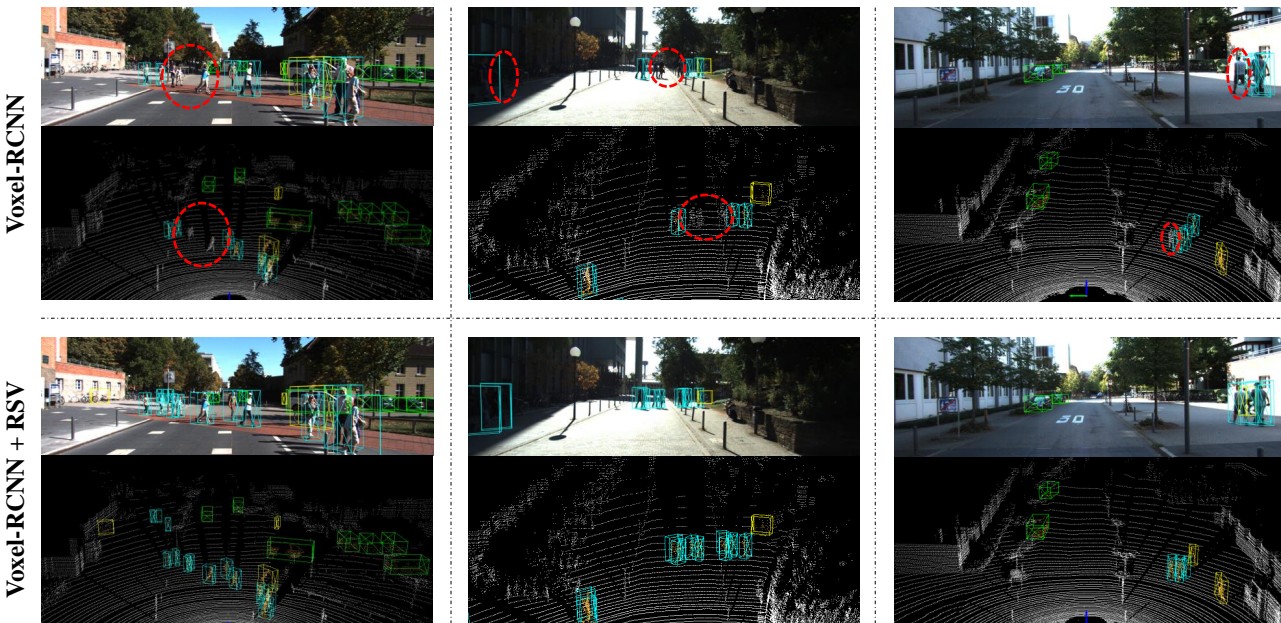

*Figure 3.* Qualitative results achieved by Voxel-RCNN and its RSV variant on the KITTI dataset. Detected objects are visualized with 3D bounding boxes in different colors: green for cars, blue for pedestrians, and yellow for cyclists. Differences of interest between the two methods are highlighted with red dashed circles.

*Table 2.* Comparison of average active voxels and GPU memory usage.

| Method | Voxel Size (m) | # Average Active Voxels | Peak GPU Memory (MB) |
|---|---|---|---|
| SECOND | [0.05, 0.05, 0.1] | 15051 | 301.39 |
| SECOND | [0.025, 0.025, 0.1] | 17672 | 982.70 |
| SECOND + RSV | [0.05, 0.05, 0.1] | 17321 | 397.32 |

*Table 3.* Comparative 3D object detection results at 0–30$m$ on the KITTI validation set. Improvements brought by RSV ($\Delta$ gain) are shown in red.

| Method | Voxel Size (m) | Pedestrian | | | Cyclist | | |
|---|---|---|---|---|---|---|---|
| | | Easy | Mod. | Hard | Easy | Mod. | Hard |
| SECOND-IoU | [0.1 , 0.1 , 0.1 ] | 53.39 | 48.52 | 45.17 | 82.12 | 69.45 | 63.50 |
| SECOND-IoU + RSV | [0.1 , 0.1 , 0.1 ] | 63.53 | 60.10 | 55.70 | 91.42 | 76.92 | 70.61 |
| $\Delta$ Gain | - | **+10.14** | **+11.58** | **+10.53** | **+9.30** | **+7.47** | **+7.11** |
| SECOND-IoU | [0.05, 0.05, 0.1 ] | 62.70 | 59.20 | 53.58 | 93.46 | 77.98 | 72.34 |
| SECOND-IoU + RSV | [0.05, 0.05, 0.1 ] | 63.98 | 60.44 | 55.30 | 93.19 | 77.38 | 71.87 |
| $\Delta$ Gain | - | **+1.28** | **+1.24** | **+1.72** | **−0.27** | **−0.60** | **−0.47** |
| PointPillars | [0.32, 0.32, 4.0 ] | 27.45 | 27.35 | 26.06 | 61.34 | 50.90 | 46.85 |
| PointPillars + RSV | [0.32, 0.32, 4.0 ] | 57.91 | 54.71 | 49.94 | 83.89 | 70.50 | 64.81 |
| $\Delta$ Gain | - | **+30.46** | **+27.36** | **+23.88** | **+22.55** | **+19.60** | **+17.96** |
| PointPillars | [0.16, 0.16, 4.0 ] | 57.07 | 54.52 | 49.81 | 81.25 | 68.82 | 64.86 |
| PointPillars + RSV | [0.16, 0.16, 4.0 ] | 59.94 | 56.01 | 50.74 | 90.59 | 75.84 | 70.15 |
| $\Delta$ Gain | - | **+2.87** | **+1.49** | **+0.93** | **+9.34** | **+7.02** | **+5.29** |
| Voxel-RCNN | [0.1 , 0.1 , 0.1 ] | 52.78 | 50.13 | 45.69 | 92.48 | 78.24 | 72.97 |
| Voxel-RCNN + RSV | [0.1 , 0.1 , 0.1 ] | 67.35 | 65.65 | 59.60 | 95.48 | 82.01 | 74.45 |
| $\Delta$ Gain | - | **+14.57** | **+15.52** | **+13.91** | **+3.00** | **+3.77** | **+1.48** |
| Voxel-RCNN | [0.05, 0.05, 0.1 ] | 67.43 | 64.09 | 58.84 | 95.38 | 81.19 | 74.02 |
| Voxel-RCNN + RSV | [0.05, 0.05, 0.1 ] | 74.32 | 70.49 | 63.38 | 95.37 | 80.99 | 74.93 |
| $\Delta$ Gain | - | **+6.89** | **+6.40** | **+4.54** | **-0.01** | **-0.20** | **+0.91** |
| VirConv-L | [0.1 , 0.1 , 0.05] | 58.38 | 52.35 | 46.32 | 90.42 | 75.41 | 69.46 |
| VirConv-L + RSV | [0.1 , 0.1 , 0.05] | 72.71 | 68.81 | 61.41 | 92.91 | 78.15 | 71.82 |
| $\Delta$ Gain | - | **+14.33** | **+16.46** | **+15.09** | **+2.49** | **+2.74** | **+2.36** |
| VirConv-L | [0.05, 0.05, 0.05] | 67.00 | 63.40 | 56.52 | 92.26 | 78.89 | 71.70 |
| VirConv-L + RSV | [0.05, 0.05, 0.05] | 71.26 | 68.41 | 60.82 | 92.51 | 79.66 | 72.43 |
| $\Delta$ Gain | - | **+4.26** | **+5.01** | **+4.30** | **+0.25** | **+0.77** | **+0.73** |

achieves significant detection gains in a much more efficient manner than naive global refinement.

To further validate the effectiveness of RSV on small objects, Table 3 reports the detection performance of pedestrians and cyclists at 0–30$m$. We observe mostly consistent gains for both categories, with notably larger improvements under coarser voxel resolutions. This suggests that RSV effectively mitigates the geometric information loss and the resulting performance degradation caused by coarse voxelization.

**Qualitative results.** Figure 3 visualizes several representative detection examples on the KITTI dataset. Without RSV, small objects (e.g., pedestrians and cyclists) are more prone to missed detections or imprecise localization. In contrast, integrating RSV produces a denser and more geometry-preserving discretization in near-range regions, thus obtaining more complete detections and tighter 3D bounding boxes for small objects. At the same time, RSV does not introduce noticeable false positives on large objects (e.g., cars).

**Results on the nuScenes validation set.** To validate the

effectiveness of RSV on different types of small objects, we compare several baseline detectors with and without RSV on the nuScenes validation set. As shown in Table 4, RSV consistently improves the overall detection performance across different baselines, increasing mAP by 0.7–4.3 points and NDS by 0.9–3.5 points. Notably, RSV brings clear gains on small objects. For example, with CenterPoint-Pillar (Yin et al., 2021) at a voxel size of [0.2m, 0.2m, 8m], RSV improves the AP of bicycles, motorcycles, and pedestrians by 6.1, 3.2, and 2.8 points, respectively. It also obtains evident

*Table 4.* Comparative 3D object detection results on the nuScenes validation set. Improvements brought by RSV ($\Delta$ Gain) are shown in red. C.V., Mot., and T.C. represent construction vehicle, motorcycle, and traffic cone, respectively.

| Method | Voxel Size (m) | mAP | NDS | Car | Truck | C.V. | Bus | Trailer | Barrier | Moto. | Byc. | Ped. | T.C. |
|---|---|---|---|---|---|---|---|---|---|---|---|---|---|
| CenterPoint-Pillar (Yin et al., 2021) | [0.4, 0.4, 8] | 42.9 | 54.9 | 79.7 | 39.6 | 11.6 | 58.6 | 30.3 | 47.3 | 36.0 | 10.0 | 66.4 | 49.1 |
| CenterPoint-Pillar + RSV | [0.4, 0.4, 8] | 47.2 | 58.4 | 79.9 | 43.2 | 12.4 | 59.7 | 31.5 | 52.9 | 43.0 | 17.4 | 74.9 | 56.5 |
| $\Delta$ Gain | - | +4.3 | +3.5 | +0.2 | +3.6 | +0.8 | +1.1 | +1.2 | +5.6 | +7.0 | +7.4 | +8.5 | +7.4 |
| CenterPoint-Pillar (Yin et al., 2021) | [0.2, 0.2, 8] | 50.5 | 60.8 | 83.2 | 50.8 | 13.2 | 61.6 | 33.1 | 60.0 | 46.9 | 18.5 | 79.2 | 58.7 |
| CenterPoint-Pillar + RSV | [0.2, 0.2, 8] | 52.1 | 62.3 | 81.9 | 49.2 | 10.7 | 62.4 | 32.9 | 63.5 | 50.1 | 24.6 | 82.0 | 63.1 |
| $\Delta$ Gain | - | +1.6 | +1.5 | -1.3 | -1.6 | -2.5 | +0.8 | -0.2 | +3.5 | +3.2 | +6.1 | +2.8 | +4.4 |
| VoxelNeXt (Chen et al., 2023) | [0.15,0.15,0.2] | 56.8 | 64.1 | 83.2 | 53.2 | 23.1 | 68.9 | 36.7 | 56.7 | 60.9 | 46.4 | 78.5 | 60.6 |
| VoxelNeXt + RSV | [0.15,0.15,0.2] | 58.5 | 65.6 | 82.7 | 53.6 | 22.6 | 68.2 | 36.6 | 62.3 | 62.8 | 48.1 | 83.0 | 65.5 |
| $\Delta$ Gain | - | +1.7 | +1.5 | -0.5 | +0.4 | +0.5 | -0.7 | -0.1 | +5.6 | +1.9 | +1.7 | +4.5 | +4.9 |
| VoxelNeXt (Chen et al., 2023) | [0.075,0.075,0.2] | 60.5 | 66.6 | 83.8 | 55.5 | 21.0 | 70.5 | 38.0 | 69.3 | 62.7 | 49.9 | 84.5 | 69.4 |
| VoxelNeXt + RSV | [0.075,0.075,0.2] | 61.2 | 67.5 | 83.3 | 56.0 | 23.3 | 67.8 | 38.7 | 68.2 | 64.9 | 51.4 | 86.4 | 72.8 |
| $\Delta$ Gain | - | +0.7 | +0.9 | -0.5 | +0.5 | +2.3 | -2.7 | +0.7 | -1.1 | +2.2 | +1.5 | +1.9 | +3.4 |

improvements on small traffic-participant categories, e.g., *traffic cone* (+4.4 points) and *barrier* (+3.5 points). Meanwhile, the performance on large-vehicle categories (e.g., car, bus, and trailer) remains comparable, with only modest fluctuations. Furthermore, larger improvements can be observed under coarser voxel resolutions, where the baseline detectors suffer more from insufficient geometric representation of small objects. Overall, these results show that RSV can generalize well beyond KITTI and remains effective across various small object categories on nuScenes.

### 4.3. Ablation Study

In this section, we conduct extensive ablation studies to validate the effectiveness of the proposed RSV. All ablation experiments are conducted on the KITTI dataset, with AP reported at the moderate difficulty level.

**Effect of Distance-Aware Features.** We first investigate how the distance-aware features introduced in Sec. 3.3 affect the detection performance. We use Voxel-RCNN (Deng et al., 2021) at a voxel size of [0.1, 0.1, 0.1]$m$ as the baseline and integrate RSV into this detector for ablation experiments. Table 5 presents the quantitative results. We observe that removing all distance-aware features consistently degrades performance, especially for pedestrians and cyclists. Introducing the normalized radial distance $r_{norm}$ or the logarithmically normalized radial distance $r_{log\_norm}$ brings noticeable gains (1.3–1.9 AP points for pedestrians, 0.8–1.2 AP points for cyclists) by providing explicit radial distance cues, while adding the min-max normalized logarithm of the scaling factor $s_{log\_norm}$ further improves small object detection by capturing local warping strength more precisely. These results demonstrate that explicitly encoding the RSV-related distance information provides complementary geometric cues to the detector, thereby enhancing small object detection.

**Comparison of Non-Uniform Discretization Strategies.** To further validate the effectiveness of our RSV in 3D object detection, we compare it with a non-uniform dis-

*Table 5.* Ablation study for distance-aware features on the KITTI validation set.

| Method | $r_{norm}$ | $r_{log\_norm}$ | $s_{log\_norm}$ | Ped. | Cyc. | Car |
|---|---|---|---|---|---|---|
| Voxel-RCNN + RSV | | | | 59.8 | 70.8 | 82.5 |
| | ✓ | | | 61.1 | 71.6 | 82.6 |
| | | ✓ | | 61.7 | 72.0 | 82.7 |
| | | | ✓ | 61.9 | 71.4 | 82.8 |
| | ✓ | | ✓ | 61.8 | 72.4 | 82.8 |
| | | ✓ | ✓ | 62.0 | 72.5 | 82.7 |
| | ✓ | ✓ | ✓ | **62.2** | **73.4** | **82.9** |

cretization scheme based on classical cylindrical coordinates. In the cylindrical discretization scheme, the point cloud is transformed into $(\rho, \phi, z)$ space and uniformly partitioned along the radial, angular, and vertical axes. For a fair comparison, we keep the same backbone, detection head, training schedule, and use comparable discretization resolutions for different schemes. Specifically, for SECOND (Yan et al., 2018), both uniform Cartesian voxelization and RSV use a voxel size of $[0.05m, 0.05m, 0.1m]$, while the corresponding cylindrical grid is set to $[\Delta\rho = 0.05m, \Delta\theta = 0.001169863\text{rad}, \Delta z = 0.1m]$. As shown in Table 6, cylindrical discretization fails to consistently improve performance and can even underperform the uniform Cartesian baseline. In contrast, the proposed RSV strategy significantly outperforms both uniform and cylindrical discretization, especially for pedestrians and cyclists. These results demonstrate that adaptively allocating finer voxels to near-field regions in a Cartesian grid is more effective for enhancing small-object detection.

*Table 6.* Ablation study for point cloud discretization strategies on the KITTI validation set.

| Method | Discretization | Ped. | Cyc. | Car |
|---|---|---|---|---|
| SECOND | Uniform | 51.1 | 66.7 | 81.5 |
| | Cylindrical | 49.9 | 59.2 | 79.6 |
| | RSV | **61.9** | **67.4** | **81.6** |

**Sensitivity to Hyperparameters.** We conduct a quantitative sensitivity analysis of the three RSV hyperparameters, namely $s_{max}$, $\beta$, and $r_{far}$, using PointPillars (Lang et al.,

*Table 7.* Ablation study of $s_{max}$ on the KITTI validation set.

| $s_{max}$ | Car | Ped. | Cyc. |
|---|---|---|---|
| 1.5 | 78.3 | 52.4 | 66.6 |
| 1.8 | 78.4 | 52.6 | 67.8 |
| 2.0 | 78.7 | 51.9 | 66.7 |
| 2.2 | 77.9 | 52.6 | 65.7 |
| 2.5 | 77.7 | 50.2 | 65.9 |

*Table 8.* Ablation study of $\beta$ on the KITTI validation set.

| $\beta$ | Car | Ped. | Cyc. |
|---|---|---|---|
| 1.1 | 78.4 | 52.7 | 67.3 |
| 1.3 | 78.2 | 52.8 | 65.1 |
| 1.5 | 78.7 | 51.9 | 66.7 |
| 1.7 | 77.3 | 52.3 | 67.1 |
| 1.9 | 78.5 | 51.7 | 66.6 |

*Table 9.* Ablation study of $r_{far}$ on the KITTI validation set.

| $r_{far}$ | Car | Ped. | Cyc. |
|---|---|---|---|
| 40 | 77.3 | 52.6 | 66.8 |
| 45 | 77.3 | 50.9 | 67.6 |
| 50 | 78.7 | 51.9 | 66.7 |
| 55 | 77.9 | 52.0 | 65.4 |
| 60 | 78.2 | 52.8 | 65.2 |

2019) with voxel size of [0.16, 0.16, 4]m. In each experiment, one hyperparameter is varied while the other two are fixed at their default values. As shown in Tables 7-9, the detection performance is relatively stable across different settings, with only minor fluctuations for all three object categories. This suggests that RSV is not sensitive to the choice of hyperparameters. Based on the overall trade-off, we use $s_{max}$=2.0, $\beta$=1.5, and $r_{far}$=50 as the default setting in all experiments.

**Generalize to 3D Multi-Object Tracking.** To verify that RSV is not limited to 3D object detection, we further apply it to a 3D multi-object tracking (MOT) framework. Specifically, we use CenterPoint-Pillar (Yin et al., 2021) with a voxel size of [0.2, 0.2, 8]m as the baseline tracker and replace its standard uniform voxelization with our RSV, while keeping all other components (e.g., backbone and heads) unchanged. Table 10 reports the 3D multi-object tracking results of CenterPoint-Pillar and its RSV variant. It is observed that RSV brings significant improvements (e.g., +24.6% in AMOTP) in overall tracking performance. Furthermore, we also note that small object tracking (e.g., for bicycle, +84.4% in AMOTA and +86.6% in AMOTP; for pedestrian, +9.6% in AMOTA and +12.8% in AMOTP) is significantly improved. These results demonstrate that our RSV can be well generalized to existing voxel-based 3D MOT frameworks.

*Table 10.* 3D multi-object tracking results on the nuScenes validation set.

| Method | AMOTP | Bicycle | | Pedestrian | |
|---|---|---|---|---|---|
| | | AMOTA | AMOTP | AMOTA | AMOTP |
| CenterPoint-Pillar | 0.647 | 0.135 | 0.119 | 0.705 | 0.592 |
| CenterPoint-Pillar + RSV | **0.806** | **0.249** | **0.222** | **0.773** | **0.668** |

**Center-based vs. Corner-based Approximation.** We further compare the default center-based approximation with a corner-based variant to examine whether a more geometrically faithful transformation improves detection performance. Specifically, the center-based variant approximates box scaling using the scaling factor evaluated at the box center, whereas the corner-based variant forward-maps the four box corners into the warped space and fits the transformed shape with an enclosing rectangle. As shown in Table 11, the corner-based variant does not yield consistent improve-

*Table 11.* Comparison of center-based and corner-based approximation.

| Method | Voxel Size (m) | Warp | Car | Ped. | Cyc. |
|---|---|---|---|---|---|
| PointPillars + RSV | [0.16, 0.16, 4] | Center | 78.7 | 51.9 | 66.7 |
| | | Corner | 77.5 | 53.2 | 65.2 |
| Voxel-RCNN + RSV | [0.05, 0.05, 0.1] | Center | 85.2 | 63.9 | 73.8 |
| | | Corner | 84.5 | 63.5 | 74.6 |

ments across different detectors and object categories. In addition, the corner-based variant requires four inverse decoding operations at inference time, which increases computational complexity and may produce looser boxes that include more background. In contrast, the center-based approximation achieves competitive performance with lower inference overhead. Therefore, we adopt the center-based approximation for box transformation.

## 5. Conclusion

In this paper, we have proposed Radial Scaling Voxelization (RSV), a simple yet effective non-uniform discretization strategy for voxel-based 3D object detectors. By applying a continuous radial scaling function to the input coordinates before voxelization, RSV increases the effective voxel density in near regions while preserving the Cartesian grid topology and keeping the far-range voxel resolution nearly unchanged. This design enables detectors to better capture the fine-grained geometry of nearby small objects, especially pedestrians and cyclists, without introducing significant memory and computational overhead. Furthermore, RSV is architecture-agnostic and can be seamlessly integrated into existing voxel-based detectors by replacing only the discretization module. Extensive experiments on the KITTI and nuScenes datasets show that RSV significantly improves small-object detection performance with only a slight increase in computational cost. These results demonstrate that RSV is an efficient solution for enhancing small-object detection and highlight the importance of non-uniform discretization in voxel-based 3D detection.

## Acknowledgements

This work was supported by the Shanghai Magnolia Talent Program Pujiang Project.

## Impact Statement

This paper presents work whose goal is to advance the field of Machine Learning. There are many potential societal consequences of our work, none of which we feel must be specifically highlighted here.

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

## A. Newton Iteration for Inverse Radial Scaling Transformation

To invert the forward transformation $(u, v, w) = (s(r)x, \ s(r)y, \ z)$, we solve for the original radius $r$ from

$$f(r) = s(r)r - \rho = 0, \qquad \rho = \sqrt{u^2 + v^2}. \tag{11}$$

The radial scaling function and its derivative are defined as:

$$s(r) = 1 + (s_{\max} - 1) \exp\left(-\beta \frac{r}{r_{\text{far}}}\right). \tag{12}$$

$$s'(r) = -(s_{\max} - 1) \frac{\beta}{r_{\text{far}}} \exp\left(-\beta \frac{r}{r_{\text{far}}}\right). \tag{13}$$

Thus,

$$f(r) = s(r)r - \rho, \quad f'(r) = s(r) + rs'(r), \tag{14}$$

The Newton update is computed as:

$$r_{\text{new}} = r - \frac{f(r)}{f'(r) + \varepsilon}, \tag{15}$$

with a small $\varepsilon$ added for numerical stability. To ensure physical validity, we clamp the radius:

$$r_{\text{new}} = \max(0, r_{\text{new}}) \tag{16}$$

The iteration terminates early once

$$|r_{\text{new}} - r| < \tau, \tag{17}$$

with $\tau$ a convergence tolerance. Empirically, the solver converges within 5–8 iterations for all transformed points. After recovering $r$, the inverse mapping follows:

- For $\rho > 0$:

$$x = \frac{r}{\rho}u, \qquad y = \frac{r}{\rho}v, \qquad z = w. \tag{18}$$

- For $\rho = 0$, the point remains at the origin:

$$x = 0, \qquad y = 0, \qquad z = w. \tag{19}$$

This completes the inverse transformation from $(u, v, w)$ back to the original LiDAR coordinate system.

# B. Pseudo-code of Radial Scaling Voxelization

This section summarizes the Radial Scaling Voxelization procedure used in our implementation, as shown in Algorithm 1. Given raw points in the LiDAR coordinates, we apply the radial scaling in the horizontal plane and voxelize the warped coordinates on a uniform grid in $(u, v, w)$ space.

---

**Algorithm 1** Radial Scaling Voxelization

---

**Input:** Point cloud $\mathcal{P} = \{(x_i, y_i, z_i)\}_{i=1}^N$ in LiDAR coordinates.
**Hyperparameters:** Scaling parameters $s_{\max}$, $\beta$, $r_{\mathrm{far}}$; voxel size $(\Delta u, \Delta v, \Delta w)$; transformed coordinate range $(u_{\min}, v_{\min}, w_{\min})$.
**Output:** Voxel grid $\mathcal{V}$ defined on a regular lattice in transformed $(u, v, w)$ space.
**for** $i = 1$ **to** $N$ **do**
    $r_i \leftarrow \sqrt{x_i^2 + y_i^2}$
    $s_i \leftarrow 1 + (s_{\max} - 1) \exp\left(-\beta \frac{r_i}{r_{\mathrm{far}}}\right)$
    $u_i \leftarrow s_i x_i, \quad v_i \leftarrow s_i y_i, \quad w_i \leftarrow z_i$
    $k_i \leftarrow \left(\left\lfloor \frac{u_i - u_{\min}}{\Delta u} \right\rfloor, \left\lfloor \frac{v_i - v_{\min}}{\Delta v} \right\rfloor, \left\lfloor \frac{w_i - w_{\min}}{\Delta w} \right\rfloor\right)$
    Assign point $(u_i, v_i, w_i)$ to voxel $\mathcal{V}[k_i]$.
**end for**
**Return:** Voxel grid $\mathcal{V}$ in the transformed $(u, v, w)$ space.
*Remark:* For a voxel located at radial distance $r$ from the origin, the corresponding LiDAR-space voxel size is
$$\Delta x(r) = \Delta u / s(r), \quad \Delta y(r) = \Delta v / s(r), \quad \Delta z = \Delta w.$$
Near-range regions with large $s(r)$ have smaller $(\Delta x, \Delta y)$, i.e., finer voxels, while far-range regions with $s(r) \approx 1$ preserve the original voxel size.

---

