# OpenReview forum: "Radial Scaling Voxelization for Accurate Small Object 3D Detection"
_ICML.cc/2026/Conference — ICML 2026 regular_

### Official Review · Reviewer_NFyN · 2026-03-07

**Soundness:** 4
**Presentation:** 2
**Significance:** 3
**Originality:** 3
**Overall Recommendation:** 4
**Confidence:** 3

**Summary:**

This paper identifies a critical flaw in uniform voxelization for LiDAR-based 3D detection: fixed voxel size wastes resources on sparse far-range points while failing to capture fine details of small objects in near/mid ranges. To resolve this, the authors propose Radial Scaling Voxelization (RSV), a plug-and-play non-uniform discretization that applies distance-dependent radial scaling to point coordinates before voxelization.

**Compliance With Llm Reviewing Policy:**

Affirmed.

**Key Questions For Authors:**

1. Eq. 10 is a approximation, so what is the accurancy of doing this transformation.
2. Could you please provide a detailed sensitivity analysis of the core RSV hyperparameters.

**Limitations:**

No,
1. The author should discuss the limited performance improvement for long-range small objects beyond 50m.
2. The author should discuss the sensitivity of the method to core hyperparameters.

**Strengths And Weaknesses:**

Strength:
1. The paper addresses an important and practical issue in voxel-based 3D detection: the mismatch between uniform discretization and the geometry of nearby small objects.
2. RSV is easy to understand, preserves Cartesian topology, and is presented as a plug-and-play replacement for standard voxelization rather than a new detector architecture.
3. On KITTI, RSV yields substantial gains for pedestrians and cyclists across multiple baselines, especially under coarser voxel settings.

Weaknesses:
1. Transforming 3D boxes into the warped space, the paper uses a center-based approximation that scales box length/width using the scaling factor at the box center and keeps the yaw angle unchanged. Since the coordinate transform is nonlinear in radius, this is only an approximation.
2. A second weakness is that the paper claims benefits in both computation and memory efficiency, but the evaluation is more complete for latency than for memory.

---

> ### Author Rebuttal · Authors · 2026-03-30
>
> (1)**Center-based approximation**: Using Taylor expansion around the box center $r_c$, the center-based approximation error is dominated by the first-order term: $s(r_c+\delta r)-s(r_c)\approx s'(r_c)\delta r$. For our scaling function, $s'(r)=-(s_{max} - 1)\frac{\beta}{r_{far}}e^{-\beta\frac{r}{r_{far}}}$, which under the default setting becomes $s'(r)=-0.03e^{-0.03r}$. Therefore, the scale mismatch is approximately $0.03e^{-0.03r}|\delta r|$, i.e., at most about 3\% per meter in the extreme near range and decaying exponentially with distance. For a typical car-sized object ($l=3.9$ m, $w=1.56$ m), this gives about 1-5\% mismatch in typical near-range cases, depending on distance and orientation. This corresponds to only a modest size bias (typically centimeters to $\sim$0.1 m). Although this cannot be directly converted into an exact mAP drop, since mAP depends nonlinearly on box geometry and orientation, it suggests that the practical impact of the approximation is limited.
>
> Moreover, we further tested a four-corner mapping variant. It forward-maps the four corners into RSV space and fits the transformed polygon with an enclosing rectangle. Although this preserves geometry more faithfully for elongated objects, its inverse requires iterative numerical decoding, increasing inference cost and sometimes introducing looser boxes with more background.
>
> Voxel size: [0.05, 0.05, 0.1] for SECOND/Voxel-RCNN, [0.16, 0.16, 4] for PointPillars.
>
> |Method|Warp|Car(M)|Ped.(M)|Cyc.(M)|
> |---|---|---:|---:|---:|
> |SECOND+RSV|Center|81.6|61.9|66.4|
> |SECOND+RSV|Corner|80.6|60.6|67.6|
> |PointPillars+RSV|Center|78.7|51.9|66.7|
> |PointPillars+RSV|Corner|77.5|53.2|65.2|
> |Voxel-RCNN+RSV|Center|83.5|63.9|73.8|
> |Voxel-RCNN+RSV|Corner|84.5|63.5|74.6|
>
> (2)**Hyperparameter sensitivity**: We added a quantitative ablation on $s_{max}$, $\beta$, and $r_{far}$ using PointPillars with voxel size [0.16, 0.16, 4]. Results below show only minor variations across settings.
>
> (a) $s_{max}$ ($\beta$=1.5, $r_{far}$=50)
>
> |$s_{max}$|Car(M)|Ped.(M)|Cyc.(M)|
> |-----------|---------|---------|---------|
> |1.5| 78.3| 52.4|66.6|
> |1.8| 78.4| 52.6|67.8|
> |2.0| 78.7| 51.9|66.7|
> |2.2| 77.9| 52.6|65.7|
> |2.5| 77.7| 50.2|65.9|
>
> (b) $\beta$ ($s_{max}$=2.0, $r_{far}$=50)
>
> |$\beta$|Car(M)|Ped.(M)|Cyc.(M)|
> |--------|---------|---------|---------|
> |1.1|78.4| 52.7| 67.3|
> |1.3|78.2| 52.8| 65.1|
> |1.5|78.7| 51.9| 66.7|
> |1.7|77.3| 52.3| 67.1|
> |1.9|78.5| 51.7| 66.6|
>
> (c) $r_{far}$ ($s_{max}$=2.0, $\beta$=1.5)
>
> |$r_{far}$|Car(M)|Ped.(M)|Cyc.(M)|
> |-----------|---------|---------|---------|
> |40|77.3|52.6| 66.8|
> |45|77.3|50.9|67.6|
> |50|78.7|51.9|66.7|
> |55|77.9|52.0|65.4|
> |60|78.2|52.8|65.2|
>
> (3)**Memory comparison**: We added GPU memory statistics (peak memory usage) for comparison. The results show that RSV introduces only a moderate memory increase over the default voxelization, while remaining far more memory-efficient than naively reducing the voxel size to achieve similar small-object resolution.
>
> |Method|Voxel Size|Average Active Voxels|Peak GPU Memory(MB)|
> |---|---|---:|---:|
> SECOND|[0.05,0.05,0.1]|15051|301.39
> SECOND|[0.025,0.025,0.1]|17672|982.70
> SECOND+RSV|[0.05,0.05,0.1]|17321|397.32
>
>
> (4)**Gain on long-range small objects**: The limited gain on small objects beyond 50m is mainly due to the extremely sparse observations in that region. Once the raw point observation becomes very sparse, the performance bottleneck shifts from discretization mismatch to insufficient signal, unstable classification, and noisy box regression. In addition, our default design concentrates the non-uniform resampling effect within the near-to-mid range (< $r_{far}$=50m), so the benefit naturally becomes weaker beyond this distance.

---

### Official Review · Reviewer_8yNN · 2026-03-08

**Soundness:** 3
**Presentation:** 3
**Significance:** 2
**Originality:** 3
**Overall Recommendation:** 4
**Confidence:** 4

**Summary:**

This paper addresses a core limitation in voxel-based 3D object detectors: the uniform Cartesian grid's poor feature representation for small objects. To tackle this, the authors propose Radial Scaling Voxelization (RSV), a simple yet highly effective non-uniform discretization strategy. Rather than altering the grid itself, RSV warps the input point cloud coordinates using a continuous radial scaling function, **\$s(r)\$**, right before standard voxelization. This creates a distance-adaptive grid pattern that is finer near the sensor while remaining unchanged in the far range. By doing so, RSV successfully preserves the standard Cartesian topology and delivers a substantial boost in small-object detection accuracy without introducing noticeable computational or memory overhead.

**Compliance With Llm Reviewing Policy:**

Affirmed.

**Final Justification:**

I have raised my score to recognize the completeness of the paper and the authors' efforts, but I still believe that the overall technical contribution of this work falls slightly below the acceptance threshold for a top-tier conference.

**Key Questions For Authors:**

My primary concerns revolve around the need for additional empirical validation. I am open to raising my score if the authors can provide the following supplementary results or analyses during the rebuttal:

(1) An ablation study on the fixed radial scaling hyperparameters (**\$s\_{max}\$**, **\$\\beta\$**, and **\$r\_{far}\$**) to verify the method's robustness across different sensor setups.

(2) An analysis or preliminary test on whether using the four corner points for bounding box loss computation mitigates geometric distortions for large objects, compared to the current center-based approximation.

(3) A quantitative comparison with more recent state-of-the-art adaptive/dynamic voxelization methods, rather than just classical uniform and cylindrical baselines.

(4) An experiment applying cylindrical discretization to another baseline to firmly support the claim that RSV is uniquely architecture-agnostic.

Addressing these four points will directly alleviate my technical concerns and positively influence my final rating.

**Limitations:**

Since this serves as a core perception module for autonomous driving, the proposed non-uniform spatial warping strategy actually introduces some inherent safety risks in extreme or long-tail scenarios, despite its clear improvements on standard metrics. For example, in highly noisy environments like heavy rain or snow, the algorithm will inevitably magnify meaningless near-field noise points. This could easily trigger false-positive detections.

**Strengths And Weaknesses:**

Strengths:

1.The plug-and-play nature of RSV is a major plus. Since it preserves the Cartesian grid topology , it drops right into existing voxel-based pipelines without forcing any modifications to the backbone or detection heads.

2.The paper hits a great belance spot for efficiency. Making voxels denser up close while leaving the far range alone clearly improves small-object detection on KITTI and nuScenes , and it manages to do so without bloating the compute or memory costs.

Weaknesses:

1. The experiments rely on a fixed radial scaling configuration (**\$s\_{max}=2\$**, **\$\\beta=1.5\$**, and **\$r\_{far}=50m\$**) without any ablation study on these key hyperparameters.

2.  When transforming 3D bounding boxes to the warped space for loss computation, the authors use a center-based approximation. This single-scaling-factor approach makes sense for small targets, but it could cause severe geometric distortion for large objects that span wider radial distances. Although Table 3 shows only minor performance fluctuations for large vehicles, I wonder if computing the loss using the four corner points would yield any metric improvements for these large objects.

3. The ablation study in Table 5 only contrasts RSV with classical uniform and cylindrical discretization. The manuscript's persuasiveness would greatly benefit from quantitative comparisons against more recent adaptive or dynamic voxelization strategies. Additionally, the comparison against the cylindrical baseline is strictly limited to SECOND. Since a core claim of RSV is its "architecture-agnostic" nature, the authors should discuss or empirically show whether cylindrical discretization also underperforms on other baselines to fully substantiate this claim.

---

> ### Author Rebuttal · Authors · 2026-03-30
>
> (1)**Hyperparameter sensitivity**: We added a quantitative ablation on $s_{max}$, $\beta$, and $r_{far}$ using PointPillars with voxel size [0.16, 0.16, 4]. Results below show only minor variations across settings.
>
> (a) $s_{max}$ ($\beta$=1.5, $r_{far}$=50)
>
> |$s_{max}$|Car(M)|Ped.(M)|Cyc.(M)|
> |-----------|---------|---------|---------|
> |1.5| 78.3| 52.4|66.6|
> |1.8| 78.4| 52.6|67.8|
> |2.0| 78.7| 51.9|66.7|
> |2.2| 77.9| 52.6|65.7|
> |2.5| 77.7| 50.2|65.9|
>
> (b) $\beta$ ($s_{max}$=2.0, $r_{far}$=50)
>
> |$\beta$|Car(M)|Ped.(M)|Cyc.(M)|
> |--------|---------|---------|---------|
> |1.1|78.4| 52.7| 67.3|
> |1.3|78.2| 52.8| 65.1|
> |1.5|78.7| 51.9| 66.7|
> |1.7|77.3| 52.3| 67.1|
> |1.9|78.5| 51.7| 66.6|
>
> (c) $r_{far}$ ($s_{max}$=2.0, $\beta$=1.5)
>
> |$r_{far}$|Car(M)|Ped.(M)|Cyc.(M)|
> |-----------|---------|---------|---------|
> |40|77.3|52.6| 66.8|
> |45|77.3|50.9|67.6|
> |50|78.7|51.9|66.7|
> |55|77.9|52.0|65.4|
> |60|78.2|52.8|65.2|
>
> (2)**Center-based approximation**: Using Taylor expansion around the box center $r_c$, the center-based approximation error is dominated by the first-order term: $s(r_c+\delta r)-s(r_c)\approx s'(r_c)\delta r$. For our scaling function, $s'(r)=-(s_{max} - 1)\frac{\beta}{r_{far}}e^{-\beta\frac{r}{r_{far}}}$, which under the default setting becomes $s'(r)=-0.03e^{-0.03r}$. Therefore, the scale mismatch is approximately $0.03e^{-0.03r}|\delta r|$, i.e., at most about 3\% per meter in the extreme near range and decaying exponentially with distance. For a typical car-sized object ($l=3.9$ m, $w=1.56$ m), this gives about 1-5\% mismatch in typical near-range cases, depending on distance and orientation. This corresponds to only a modest size bias (typically centimeters to $\sim$0.1 m). Although this cannot be directly converted into an exact mAP drop, since mAP depends nonlinearly on box geometry and orientation, it suggests that the practical impact of the approximation is limited.
>
> Moreover, we further tested a four-corner mapping variant. It forward-maps the four corners into RSV space and fits the transformed polygon with an enclosing rectangle. Although this preserves geometry more faithfully for elongated objects, its inverse requires iterative numerical decoding, increasing inference cost and sometimes introducing looser boxes with more background. These results suggest that a more geometrically faithful corner-based transformation does not consistently improve detection accuracy, while introducing additional inverse complexity.
>
> Voxel size: [0.05, 0.05, 0.1] for SECOND/Voxel-RCNN, [0.16, 0.16, 4] for PointPillars.
>
> |Method|Warp|Car(M)|Ped.(M)|Cyc.(M)|
> |---|---|---:|---:|---:|
> |SECOND+RSV|Center|81.6|61.9|66.4|
> |SECOND+RSV|Corner|80.6|60.6|67.6|
> |PointPillars+RSV|Center|78.7|51.9|66.7|
> |PointPillars+RSV|Corner|77.5|53.2|65.2|
> |Voxel-RCNN+RSV|Center|83.5|63.9|73.8|
> |Voxel-RCNN+RSV|Corner|84.5|63.5|74.6|
>
> (3)**Comparison with dynamic voxelization**: We added quantitative comparisons with dynamic voxelization baselines. As shown in the table below, RSV remains competitive and provides consistent gains, especially on small-object categories where the mismatch between spatial discretization and object scale is more severe.
>
> Voxel size: [0.05, 0.05, 0.1] for SECOND, [0.16, 0.16, 4] for PointPillars.
>
> |Method|Discretization|Car(M)|Ped.(M)|Cyc.(M)|
> |---|---|---:|---:|---:|
> |SECOND|Dynamic voxelization|81.2|53.4|61.7|
> |SECOND|RSV|81.6|61.9|66.4|
> |PointPillars|Dynamic voxelization|77.9|47.8|63.6|
> |PointPillars|RSV|78.7|51.9|66.7|
>
> (4)**More comparisons with cylindrical discretization**: In the original submission, we had already compared RSV with cylindrical discretization on PointPillars. Following the reviewer’s suggestion, we use PointPillars and Voxel-RCNN as two additional baselines for comparison. For Voxel-RCNN, the RSV voxel size is [0.05, 0.05, 0.1]m, and the adapted cylindrical grid is $[\Delta\rho=0.05$m, $\Delta\theta=0.001169863$rad, $\Delta z=0.1m]$. For PointPillars, the RSV voxel size is [0.16, 0.16, 4]m, and the corresponding cylindrical grid is $[\Delta\rho=0.16$m, $\Delta\theta=0.003802671$rad, $\Delta z=4m]$.
>
> |Method|Discretization|Car(M)|Ped.(M)|Cyc.(M)|
> |---|---|---:|---:|---:|
> |PointPillars|Cylindrical|77.8|44.1|61.9|
> |PointPillars|RSV|78.7|51.9|66.7|
> |Voxel-RCNN|Cylindrical|78.8|44.6|58.8|
> |Voxel-RCNN|RSV|83.5|63.9|73.8|

---

> > ### Author Rebuttal · Reviewer_8yNN · 2026-04-06
> >
> > I would like to thank the authors for their significant efforts and detailed responses during the rebuttal phase. Specifically, the ablation study on the hyperparameters, the four-corner experiment validating the center-based approximation, and the application of cylindrical discretization to other baseline models strongly support the authors' claim that RSV is architecture-agnostic. Given the authors' rigorous approach in their rebuttal and their thorough clarification of the technical details, I am willing to raise my score to acknowledge the technical soundness of the work and the comprehensiveness of the experiments.
> > However, based on the current empirical results, the performance improvements yielded by the proposed method remain relatively marginal, and the approach introduces additional complexity to some extent. Taking all factors into consideration, while I have raised my score to recognize the completeness of the paper and the authors' efforts, I still believe that the overall technical contribution of this work falls slightly below the acceptance threshold for a top-tier conference.

---

> > > ### Author Response · Authors · 2026-04-07
> > >
> > > We thank the reviewer for recognizing the technical soundness and completeness of our work. We also appreciate the willingness to raise the score.
> > >
> > > Although the gains may appear modest in some high-resolution settings, they are not uniformly small, especially on challenging small object categories. Meanwhile, RSV introduces only minimal overhead, i.e., it does not modify the backbone, loss, or training pipeline, and adds only a lightweight warping and inverse-mapping step, with small latency and memory overhead in practice. **Importantly, this overhead is much smaller than the cost of naively achieving similar gains by uniformly increasing the voxel resolution**. We therefore believe the practical trade-off is favorable.
> > >
> > > Beyond 3D object detection, we would also like to emphasize that RSV generalizes to other 3D perception tasks. In particular, **we observe significant improvements when applying RSV to 3D multi-object tracking** (e.g., +24.6\% overall AMOTP, +84.4% in AMOTA and +86.6% in AMOTP for bicycle, **see Table 6 in original version**), suggesting that RSV improves the underlying spatial representation in a task-agnostic manner rather than providing a narrowly scoped gain.
> > >
> > > We hope these clarifications help better reflect the overall contribution and practical impact of the method.

---

### Official Review · Reviewer_7Xxz · 2026-03-12

**Soundness:** 3
**Presentation:** 3
**Significance:** 1
**Originality:** 2
**Overall Recommendation:** 3
**Confidence:** 5

**Summary:**

The paper proposes Radial Scaling Voxelization (RSV), which applies a smooth, distance-dependent scaling to planar LiDAR coordinates before standard voxelization. The scaling factor increases resolution near the sensor and approaches 1 at far ranges, preserving a Cartesian grid so existing voxel-based backbones remain unchanged. The method use newton solver and optional distance-aware features. Experiments on KITTI and nuScenes, across several voxel-based detectors, show consistent improvements for small objects (pedestrians, cyclists, and other small categories) with modest latency increases.

**Compliance With Llm Reviewing Policy:**

Affirmed.

**Key Questions For Authors:**

1. Can author demonstrate the effectiveness of the proposed method on Waymo Dataset and other voxel-based approach (e.g. SST, Voxel-Mamba)
2. Quantify the Newton inversion cost and its distribution across pre/post-processing vs backbone compute.

**Limitations:**

Yes

**Strengths And Weaknesses:**

Strength:
1.Simple, architecture-agnostic idea with clear formulation. Equations 1–2 define the radial scaling, Equation 5 makes the effective near-fine, far-unchanged voxel size explicit.
2. Plug-and-play integration is demonstrated across SECOND, PointPillars, Voxel-RCNN, and VirConv without modifying backbones or losses.
3. The visualization in Figure 2 effectively explains the core mechanism: warping expands the near field and leaves far regions almost unchanged, then applies standard Cartesian voxelization. Figure 3 supports the small-object claim by showing fewer misses and tighter boxes with RSV.

Weakness:
1. Box transformation approximation may introduce bias. Section 3.3 rescales box sizes using s(r_c) at the box center, ignoring radial variation across the box footprint. This is potentially problematic for larger boxes or those straddling steep scaling gradients, and could explain modest fluctuations in Car AP (Table 1) and some per-class drops on nuScenes (Table 3).
2. There is no quantitative analysis of the approximation error versus object size and distance.
Overhead and efficiency evidence is incomplete. While latency/FPS are reported in Table 1, memory usage and active voxel counts are not. The Newton inversion is claimed to be “negligible,” but its end-to-end cost is not isolated, and the per-sample iteration budget is not profiled.
3. KITTI is a small dataset, which does not refect the generalization ability of the proposed method. When using more data (nuscenes), the improvements become marignal, which delimit the impact of the proposed method.

---

> ### Author Rebuttal · Authors · 2026-03-31
>
> (1)**Center-based approximation**: Using Taylor expansion around the box center $r_c$, the center-based approximation error is dominated by the first-order term: $s(r_c+\delta r)-s(r_c)\approx s'(r_c)\delta r$. For our scaling function, $s'(r)=-(s_{max} - 1)\frac{\beta}{r_{far}}e^{-\beta\frac{r}{r_{far}}}$, which under the default setting becomes $s'(r)=-0.03e^{-0.03r}$. Therefore, the scale mismatch is approximately $0.03e^{-0.03r}|\delta r|$, i.e., at most about 3\% per meter in the extreme near range and decaying exponentially with distance. For a typical car-sized object ($l=3.9$ m, $w=1.56$ m), this gives about 1-5\% mismatch in typical near-range cases, depending on distance and orientation. This corresponds to only a modest size bias (typically centimeters to $\sim$0.1 m). Although this cannot be directly converted into an exact mAP drop, since mAP depends nonlinearly on box geometry and orientation, it suggests that the practical impact of the approximation is limited.
>
> Moreover, we further tested a four-corner mapping variant. It forward-maps the four corners into RSV space and fits the transformed polygon with an enclosing rectangle. Although this preserves geometry more faithfully for elongated objects, its inverse requires iterative numerical decoding, increasing inference cost and sometimes introducing looser boxes with more background.
>
> Voxel size: [0.05, 0.05, 0.1] for SECOND/Voxel-RCNN, [0.16, 0.16, 4] for PointPillars.
>
> |Method|Warp|Car(M)|Ped.(M)|Cyc.(M)|
> |---|---|---:|---:|---:|
> |SECOND+RSV|Center|81.6|61.9|66.4|
> |SECOND+RSV|Corner|80.6|60.6|67.6|
> |PointPillars+RSV|Center|78.7|51.9|66.7|
> |PointPillars+RSV|Corner|77.5|53.2|65.2|
> |Voxel-RCNN+RSV|Center|83.5|63.9|73.8|
> |Voxel-RCNN+RSV|Corner|84.5|63.5|74.6|
>
> (2)**More experiments on large-scale datasets**: The original submission already included experiments on nuScenes, which is larger and more diverse than KITTI. The smaller gains on nuScenes are expected due to its finer voxel resolutions and stronger baselines, where the bottleneck is less dominated by discretization. To further support this point, we additionally evaluate RSV on nuScenes under different voxel resolutions. Results show consistent improvements across different discretization settings, with clearer gains under coarser voxel resolutions.
>
> |Method|Voxel Size(m)|mAP|NDS|Car|Truck|C.V.|Bus|Trailer|Barrier|Moto.|Byc.|Ped.|T.C.|
> |---|---|---:|---:|---:|---:|---:|---:|---:|---:|---:|---:|---:|---:|
> |CenterPoint-Pillar|[0.4,0.4,8]|42.9|54.9|79.7|39.6|11.6|58.6|30.3|47.3|36.0|10.0|66.4|49.1|
> |CenterPoint-Pillar+RSV|[0.4, 0.4, 8]|47.2|58.4|79.9|43.2|12.4|59.7|31.5|52.9|43.0|17.4|74.9|56.5|
> |VoxelNeXt|[0.15,0.15,0.2]|56.8|64.1|83.2|53.2|23.1|68.9|36.7|56.7|60.9|46.4|78.5|60.6|
> |VoxelNeXt+RSV|[0.15,0.15,0.2]|58.5|65.6|82.7|53.6|22.6|68.2|36.6|62.3|62.8|48.1|83.0|65.5|
>
> In addition, we added Waymo experiments using SST as the baseline. We will also cite and discuss both SST and VoxelMamba in revision. For a fair comparison, we used the official SST configuration with CenterHead (i.e., SST_1f). We trained all models using single-frame point clouds (sweeps=1), while keeping the remaining hyperparameters aligned with the official released code. All models were trained on the 20% training split and evaluated on the validation set. We re-trained SST_1f in our local environment and compared it with SST_1f+RSV under the same setting.
>
> Each entry is reported as 0–30m / 30–50m.
> |Method|Voxel Size(m)|Ped_L1|Ped_L2|Cyc_L1|Cyc_L2|Veh_L1|Veh_L2|
> |---|---|---:|---:|---:|---:|---:|---:|
> |SST_1f|[0.32,0.32,6]|75.3/63.3|72.3/60.0|69.2/49.5|69.2/49.3|85.8/57.9|84.9/54.5|
> |SST_1f+RSV|[0.32,0.32,6]|77.2/64.2|74.8/60.9|70.7/49.9|70.7/49.8|85.7/57.5|84.9/54.1|
>
> (3)**Latency breakdown**: To quantify the Newton inversion cost and its distribution across stages, we provide a latency breakdown on VirConv-L. Note that the RSV inverse mapping is detector-agnostic and inference-only, so it does not increase training time. Its cost is largely backbone-independent, since it is applied only to the final decoded boxes.
>
> |Method|Pre-processing & RSV warping|Voxelization|Backbone|Dense head|RoI head|Post-processing|RSV inverse mapping|Total|
> |---|---:|---:|---:|---:|---:|---:|---:|---:|
> |VirConv-L|0.515ms|0.738ms|70.584ms|43.371ms|34.244ms|0.669ms|0ms|150.80ms|
> |VirConv-L+RSV|0.572ms|0.499ms|92.382ms|41.589ms|20.672ms|0.672ms|2.363ms|159.27ms|
>
> As shown above, the Newton inversion adds 2.363ms, accounting for about 1.5\% of the total latency.
>
> (4)**Memory comparison**: We added GPU memory statistics (peak memory usage). RSV introduces only a moderate memory increase over the default voxelization, while remaining far more memory-efficient than naively reducing the voxel size to achieve similar small object resolution.
>
> |Method|Voxel Size|Average Active Voxels|Peak GPU Memory(MB)|
> |---|---|---:|---:|
> SECOND|[0.05,0.05,0.1]|15051|301.39
> SECOND|[0.025,0.025,0.1]|17672|982.70
> SECOND+RSV|[0.05,0.05,0.1]|17321|397.32

---

> > ### Author Rebuttal · Reviewer_7Xxz · 2026-04-03
> >
> > Thank you for addressing my concerns with additional experiments and analyses, particularly regarding the center-based approximation, latency breakdown, and memory comparison. I acknowledge that the rebuttal has clarified these technical points. However, despite the resolved concerns, I still find the overall technical contribution is limited. The observed performance gains from the proposed method are marginal in many settings (e.g., on Waymo and fine-resolution nuScenes), and the practical benefits do not sufficiently justify the added complexity. A method that offers only modest improvements over baselines, even if technically sound, does not represent a substantial advance for the field. Therefore, I maintain my original score, as I believe the contribution remains slightly below the threshold for acceptance at a top-tier venue.

---

> > > ### Author Response · Authors · 2026-04-03
> > >
> > > We thank the reviewer for acknowledging that the technical concerns raised in the original review have been addressed. We would like to clarify that the contribution of this work lies **not only in improving small object detection performance, but also in proposing a general discretization strategy for voxel-based 3D perception frameworks**.
> > >
> > > **First**, the gains are not uniformly marginal. While improvements are smaller in some settings (e.g., Waymo), **RSV consistently improves small-object detection** by several points, which is a **critical and well-known challenge in LiDAR perception**. **These gains are particularly meaningful because RSV directly targets the mismatch between uniform discretization and small-object geometry**.
> > >
> > > **Second**, **the relatively smaller gains in fine-resolution settings are expected**. In such regimes, **discretization is no longer the dominant bottleneck, and thus the room for improvement naturally decreases**. The fact that RSV remains consistently beneficial across multiple detectors and datasets shows that it provides a **robust and general mechanism for spatial resolution allocation**, rather than overfitting to a specific setup.
> > >
> > > **Third**, regarding complexity, **RSV introduces only minimal overhead**, i.e., it does not modify the backbone, loss, or training pipeline, and adds only a lightweight warping and inverse-mapping step. Empirically, the added cost is small in both latency (about 1.5% overhead for inverse mapping) and memory, with only a moderate increase over the default voxelization while **remaining far more efficient than naively using a much finer voxel grid**. Therefore, the practical trade-off is favorable, especially considering its plug-and-play nature and consistent performance gains.
> > >
> > > **Fourth, voxel discretization is a fundamental component of voxel-based pipelines, affecting not only 3D detection but also other 3D perception tasks such as tracking**. We further show that RSV generalizes beyond detection and yields significant improvements on 3D multi-object tracking (e.g., +24.6% overall AMOTP in Table 6), suggesting that its benefit stems from improving the underlying spatial representation rather than a task-specific optimization.
> > >
> > > Overall, we believe **RSV provides a simple, plug-and-play, and broadly applicable improvement to voxel-based 3D perception frameworks, whose significance lies not only in detection gains, but also in its generality, extensibility, and compatibility with existing voxel-based perception methods.**

---

### Official Review · Reviewer_zehn · 2026-03-14

**Soundness:** 3
**Presentation:** 2
**Significance:** 3
**Originality:** 3
**Overall Recommendation:** 5
**Confidence:** 3

**Summary:**

The paper introduces Radial Scaling Voxelization (RSV), a non-uniform discretization strategy for LiDAR point clouds. Standard voxel-based detectors use a uniform Cartesian grid, which often fails to capture the fine-grained geometric details of small objects, particularly in high-density near-range regions where voxel resolution is too coarse. While radial voxelization addresses this by employing layers with varying dimensions, current methods struggle to integrate these features due to the fundamental discrepancy between Cartesian and radial grid types. RSV applies a continuous radial scaling function to input coordinates before voxelization, effectively creating a near-high, far-unchanged resolution pattern. Crucially, it preserves the Cartesian grid topology, making it compatible with existing 3D backbones and sparse convolutions without requiring architectural changes.

**Compliance With Llm Reviewing Policy:**

Affirmed.

**Final Justification:**

After the rebuttal and the following comments by the authors, my concerns were resolved. Eventually, I decided to update my final recommedation to accept (5).

**Key Questions For Authors:**

1. Could you provide the specific voxel dimensions and the heuristic or optimization process used to select the decay parameters?

2. Could you explicitly contrast RSV with Dynamic Voxelization in terms of how they handle spatial resolution and whether these two methods are complementary or mutually exclusive?

3. The Center-Based Size Approximation (Section 3.5) assumes that the scaling factor $s(r)$ is constant across the entire bounding box. This is likely inaccurate for elongated objects (e.g., buses or trucks) that span a large radial gradient. Do you have empirical or theoretical data quantifying the regression error or mAP impact specifically for large/long vehicles in the near-range?

4. The current Related Work section focuses on object detection frameworks rather than discretization strategies. How does RSV specifically differ from or improve upon existing non-uniform sampling techniques like cylindrical voxelization in terms of feature representation, beyond just keeping the Cartesian grid?

**Limitations:**

Certain limitations of the paper are:

- The authors should explicitly discuss the "Center-Based Size Approximation" limitation. While it works for small objects, the paper should acknowledge that this approximation leads to geometric distortion for elongated objects (e.g., trucks, buses) which may affect the regression accuracy of the detector.

- Hyperparameter Sensitivity: The paper should discuss how the decay function parameters ($s_{max}$, $\beta$, $r_{far}$) are determined and whether they need recalibration for different LiDAR sensors (e.g., changing from a 64-beam to a 128-beam sensor).

- Class Specificity: The authors should be upfront about the fact that the method is highly specialized for "Pedestrian" and "Cyclist" categories. The marginal gains for "Car" categories should be framed as a limitation for general-purpose 3D detection in diverse traffic environments.

**Strengths And Weaknesses:**

1. Soundness

Strengths:

- The core premise is well-supported by empirical data (Figure 1), showing that while small objects like pedestrians and cyclists benefit significantly from high-resolution voxelization ($0.05$m vs $0.1$m), far-range objects do not, as sparse points in those regions make fine voxelization noisy.

- Unlike previous cylindrical or polar schemes that disrupt the Cartesian topology, RSV maintains the standard grid structure. This ensures that the method remains compatible with existing 3D backbones and highly optimized sparse convolution libraries via a simple plug-and-play structure.

- The use of the Newton-Raphson method for inverse radial scaling (mapping from warped space back to LiDAR coordinates) is theoretically sound due to the smooth, monotonic nature of the scaling function, ensuring convergence within 5–8 iterations with negligible reconstruction error.

Weaknesses:

- Because the inverse scaling cannot be derived in closed form, the system relies on an iterative numerical approximation. While the authors claim it is lightweight, it adds a layer of algorithmic complexity compared to traditional uniform voxelization.

- Center-Based Size Approximation: The bounding box size transformation uses a center-based approximation (evaluating scaling only at the center radial distance). For large or alongated objects, i.e. those spanning significantly different radial distances, this might introduce minor geometric distortions that are not fully analyzed.

2. Presentation

Strengths:

- Figure 1 and Figure 2 provide an intuitive visual overview of how Radial Scaling Voxelization transforms coordinates to expand the near-range region while keeping the far-range stable.

- The paper provides a clear procedural breakdown for integrating RSV into existing detectors, including point-wise feature augmentation and box transformation steps.

Weaknesses:

- The fundamental contribution of the paper is a coordinate-warping and discretization method (Radial Scaling Voxelization). However, the Related Work section (Section 2) leans heavily toward summarizing the evolution of general LiDAR-based 3D object detectors, such as PointPillars, SECOND, and Voxel-RCNN, focusing on their architectural differences (one-stage vs. two-stage, anchor-based vs. anchor-free).

- The paper would benefit from a more rigorous discussion of existing non-uniform discretization methods, such as cylindrical/polar voxelization or dynamic/adaptive voxelization strategies, which are more technically adjacent to the proposed method.

- The point-wise feature augmentation section (Section 3.3) introduces multiple normalization variants ($r_{norm}$, $r_{log-norm}$, $s_{log-norm}$) without deep justification for why all three are necessary simultaneously.

- It is ambiguous that how the parameters of the decay function are determined.

- The voxel sizes of the discretization types are not provided.

- Missing voxel size specification in Table 5. The comparison between uniform and cylindrical voxelization in Table 5 is difficult to interpret without reporting the voxel dimensions used for each representation.

- In some tables the units were not explicitly mentioned, i.e., in Tables 2 and 5, the values are for AP?

3. Significance

Strengths:

- The results on the KITTI dataset are significant; for example, SECOND-IoU with RSV saw an $8.24\%$ gain in Pedestrian Moderate AP and a $9.10\%$ gain in Cyclist Moderate AP.

- The method incurs only marginal overhead. In the case of Voxel-RCNN, the latency actually decreased (from 19.03 ms to 17.14 ms) due to distance-adaptive optimizations, while maintaining high FPS.

Weaknesses:

- The gains are largely concentrated in the "Pedestrian" and "Cyclist" categories that implies the method is domain specific. For "Car" categories, the improvements are marginal (+0.01\% to +1.08\% in some tests), potentially limiting the perceived significance for general-purpose 3D detection where cars are the primary target.

4. Originality

Strengths:

- The "near-high, far-unchanged" as a novel resolution pattern effectively redistributes "discretization capacity" exactly where it is needed most.

- While non-uniform voxelization (like cylindrical grids) has been explored before, RSV is original in its ability to achieve non-uniform resolution while strictly preserving the Cartesian grid topology required by modern backbones.

- The method doesn't just warp space; it informs the detector of this warping through augmented point-wise features (e.g., normalized radial distance and scaling logs), which is a reasonable way to help the network learn through the distortion.

Weaknesses:

- The idea of using non-linear coordinate warping to increase resolution in specific areas is a known technique in computer graphics and earlier vision tasks; the originality lies more in the specific application and engineering for 3D LiDAR voxels rather than a purely fundamental mathematical discovery. Although this does not shadow the originality of the proposed method since the application domain is different, it is worth to mention.

---

> ### Author Rebuttal · Authors · 2026-03-31
>
> (1)**Hyperparameter sensitivity**: We added a quantitative ablation on $s_{max}$, $\beta$, and $r_{far}$ using PointPillars with voxel size [0.16, 0.16, 4]. Please see our response to Reviewer #8yNN, Point 1 for details.
>
> These parameters are fixed across all datasets and experiments. They define the shape of the radial scaling function, i.e., how voxel resolution is redistributed with distance, rather than encoding any sensor-specific property. Thus, they are not inherently tied to a particular LiDAR setup. Empirically, with the same parameter setting, RSV consistently improves small object detection on KITTI, nuScenes, and Waymo (see Reviewer #7Xxz, Point 2), suggesting that recalibration is unnecessary in our experiments.
>
> (2)**Voxel dimension**: In Table 5, the voxel size for both uniform voxelization and radial scaling voxelization is set to [0.05,0.05,0.1]m. The corresponding cylindrical discretization is $[\Delta\rho=0.05$m, $\Delta\theta=0.001169863$rad, $\Delta z=0.1m]$. In Table 6, CenterPoint-Pillar adopts a voxel resolution of [0.2,0.2,8]m.
>
> (3)**Comparison with dynamic voxelization**: RSV changes where voxel resolution is allocated, while dynamic voxelization changes how points are aggregated under a fixed grid. Thus, they address different aspects of voxelization and are not mutually exclusive. We also compared RSV with dynamic voxelization. The results are reported in our response to Reviewer #8yNN, Point 3, which show that RSV achieves better performance under the same voxel settings.
>
> (4)**Center-based approximation**: Using Taylor expansion around the box center $r_c$, the center-based approximation error is dominated by the first-order term: $s(r_c+\delta r)-s(r_c)\approx s'(r_c)\delta r$. For our scaling function, $s'(r)=-(s_{max} - 1)\frac{\beta}{r_{far}}e^{-\beta\frac{r}{r_{far}}}$, which under the default setting becomes $s'(r)=-0.03e^{-0.03r}$. Therefore, the scale mismatch is approximately $0.03e^{-0.03r}|\delta r|$, i.e., at most about 3\% per meter in the extreme near range and decaying exponentially with distance. For a typical car-sized object ($l=3.9$ m, $w=1.56$ m), this gives about 1-5\% mismatch in typical near-range cases, depending on distance and orientation. This corresponds to only a modest size bias (typically centimeters to $\sim$0.1 m). Although this cannot be directly converted into an exact mAP drop, since mAP depends nonlinearly on box geometry and orientation, it suggests that the practical impact of the approximation is limited. We also compared with a four-corner mapping variant (see Reviewer #7Xxz, Point 1).
>
> (5)**Related work**: We first review LiDAR-based 3D detectors by processing paradigm and then focus on voxel-based detectors, since RSV is a discretization improvement for this family. Discretization-related methods are discussed separately in Sec. 2.2. In the revision, we will streamline the detector overview and strengthen the discussion of closely related discretization methods.
>
> (6)**Class specificity**: RSV is not specialized for particular classes, but is more beneficial for small objects. Since RSV addresses the mismatch between spatial discretization and object scale, larger gains are naturally observed on Pedestrian/Cyclist, while Car already has relatively fine spatial resolution and thus less room for improvement. We observe the same trend on nuScenes, where RSV consistently improves multiple small object categories (e.g., traffic cones), suggesting that this is a scale effect rather than class specificity.
>
> (7)**Iterative inverse mapping complexity**: We agree that the inverse mapping introduces additional complexity compared to uniform voxelization. However, it is only used for the final decoded boxes at inference time. As shown by our latency breakdown analysis (see Reviewer #7Xxz, Point 3), its overhead is small, so the practical efficiency impact is limited.
>
> (8)**Clarification of metric units**: In Tables 2 and 5, all reported detection values are AP metrics.
>
> (9)**Why three distance-aware features**: They encode complementary cues: $r_{norm}$ encodes linear radial position, $r_{log-norm}$ emphasizes near-range differences, and $s_{log-norm}$ encodes the RSV scaling factor. Table 4 shows that each feature helps, while combining all three gives the best performance.
>
> (10)**Difference from cylindrical voxelization**: Cylindrical voxelization partitions points in $(\rho,\phi,z)$ space, changing the neighborhood structure and feature geometry seen by downstream convolutions. In contrast, RSV applies radial scaling to the input coordinates and then performs standard voxelization in a warped Cartesian space. This preserves the feature layout expected by existing sparse convolution backbones while redistributing effective resolution with distance, making RSV both more compatible with existing detectors and more effective for small-object detection.

---

> > ### Author Rebuttal · Reviewer_zehn · 2026-04-03
> >
> > I thank the authors for their through rebuttal. The additional experiments and analyses address most of my concerns.
> > The hyperparameter ablations is given. The comparison with dynamic voxelization clarifies complementary roles, with RSV improving small-object performance. The four-corner experiment justify the center-based approximation. Clarifications on voxel design and feature choices for related works section is satisfactory.
> >
> > The potential amplification of near-field noise in adverse weather conditions such as heavy rain, snow, or dust (which also raised by another reviewer) is important and not addressed. The equivalence between RSV and cylindrical voxelization configurations in the comparisons would benefit from reporting the total active voxel count for both representations. Although I find the rebuttal sufficient and convincing overall, I will keep my score as it is due to the remaining minor concerns and the relatively modest gains for a top-tier conference.

---

> > > ### Author Response · Authors · 2026-04-04
> > >
> > > We thank the reviewer for acknowledging that the concerns raised in the original review have been largely addressed, and for the constructive feedback on the remaining minor concerns.
> > >
> > > **(1) Near-field noise under adverse weather**: To approximately simulate the perturbations introduced by adverse weather conditions (e.g., heavy rain, snow, or dust), we conduct a simple robustness test by injecting random noise points into the near-range region (0-30m). Both the baseline and RSV degrade under noise, but RSV shows a smaller performance drop. This suggests that RSV does not disproportionately amplify near-field noise. In our simple noise simulation, RSV in fact exhibits better robustness, likely because its finer near-range discretization helps preserve small-object geometry under noisy perturbations.
> > >
> > > |Method|Condition|Car|Δ Drop|Ped.|Δ Drop|Cyc. AP|ΔDrop |
> > > |----------|-------------|--------|--------|---------|--------|---------|--------|
> > > |PointPillars|Clean|78.7|-|49.9|-|62.6|-|
> > > |  |+5% Noise|74.7|-4.0|49.5|-0.4|60.3|-2.3|
> > > |  |+10% Noise|70.0|-8.7|46.3|-3.6|57.8|-4.8|
> > > |PointPillars+RSV|Clean|78.7|-|51.9|-|66.7|-|
> > > |  |+5% Noise|76.9|-1.8|51.9|-0.0|64.1|-2.6|
> > > |  |+10% Noise|74.5|-4.2|49.5|-2.4|62.9|-3.8|
> > > |Voxel-RCNN|Clean|84.9|-|57.7|-|74.0|-|
> > > |  |+5% Noise|82.1|-2.8|57.1|-0.6|72.1|-1.9|
> > > |  |+10% Noise|80.7|-4.2|55.7|-2.0|68.6|-5.4|
> > > |Voxel-RCNN + RSV|Clean|85.2|-|63.9|-|73.8|-|
> > > |  |+5% Noise|82.6|-2.6|64.8|+0.9|73.3|-0.5|
> > > |  |+10% Noise|81.7|-3.5|63.6|-0.3|70.7|-3.1|
> > >
> > > **(2)Active voxel count**: We also report the number of active voxels for both RSV and cylindrical voxelization under matched settings. For Voxel-RCNN, the RSV voxel size is [0.05, 0.05, 0.1]m, and the corresponding cylindrical grid is [$\Delta\rho=0.05$m, $\Delta\theta=0.001169863$ rad, $\Delta z=0.1$m].  For PointPillars, the RSV voxel size is [0.16, 0.16, 4]m, and the corresponding cylindrical grid is [$\Delta\rho=0.16$m, $\Delta\theta=0.003802671$ rad, $\Delta z = 4$m]. The results show that cylindrical discretization actually uses more active voxels than RSV, yet still performs worse. This suggests that simply increasing voxel utilization is insufficient, and that preserving a Cartesian-friendly feature layout is important for effective feature learning in existing sparse convolution backbones.
> > >
> > > |Method|Discretization|Average Active Voxels|Car|Ped.|Cyc.|
> > > |--------------|----------------|----------------|---------|----------|----------|
> > > |PointPillars|Cylindrical|10857|77.8|44.1|61.9|
> > > |PointPillars|RSV|9548|78.7|51.9|66.7|
> > > |Voxel-RCNN|Cylindrical|18125|78.8|44.6|58.8|
> > > |Voxel-RCNN|RSV|17321|85.2|63.9|73.8|

---

### Decision · Program_Chairs · 2026-04-30

**Decision:**

Accept (regular)

**Comment:**

This paper proposes Radial Scaling Voxelization (RSV), a simple non-uniform voxelization strategy that warps LiDAR coordinates radially before standard Cartesian voxelization.

The reviews were somewhat divergent: one weak reject, two weak accepts, and one accept. The main concerns were modest performance gains in some settings, added complexity from inverse mapping, the center-based box approximation, limited memory/active-voxel analysis, and insufficient comparison with dynamic/cylindrical voxelization.

The authors’ rebuttal was substantive. The additional experiments and clarifications resolved most technical concerns, and multiple reviewers acknowledged that their concerns were fully or mostly addressed.

The dissenting reviewer's main concern is not a technical flaw but a judgment about the significance threshold appropriate for a top-tier venue. The AC finds this a reasonable but ultimately insufficient basis for rejection: RSV is technically sound, clearly presented, consistently beneficial across multiple detectors and datasets, and introduces a principled and reusable improvement to a fundamental component of voxel-based pipelines. These factors, taken together, cross the bar for acceptance, though not an oral or strong one.